# Temporal-Difference Variational Continual Learning

**Luckeciano C. Melo**[*1,2]     **Alessandro Abate**[†2]     **Yarin Gal**[†1]
[1] OATML, University of Oxford     [2] OXCAV, University of Oxford

## Abstract

Machine Learning models in real-world applications must continuously learn new tasks to adapt to shifts in the data-generating distribution. Yet, for Continual Learning (CL), models often struggle to balance learning new tasks (plasticity) with retaining previous knowledge (memory stability). Consequently, they are susceptible to Catastrophic Forgetting, which degrades performance and undermines the reliability of deployed systems. In the Bayesian CL literature, variational methods tackle this challenge by employing a learning objective that recursively updates the posterior distribution while constraining it to stay close to its previous estimate. Nonetheless, we argue that these methods may be ineffective due to compounding approximation errors over successive recursions. To mitigate this, we propose new learning objectives that integrate the regularization effects of multiple previous posterior estimations, preventing individual errors from dominating future posterior updates and compounding over time. We reveal insightful connections between these objectives and Temporal-Difference methods, a popular learning mechanism in Reinforcement Learning and Neuroscience. Experiments on challenging CL benchmarks show that our approach effectively mitigates Catastrophic Forgetting, outperforming strong Variational CL methods.

## 1   Introduction

A fundamental aspect of robust Machine Learning (ML) models is to learn from non-stationary sequential data. In this scenario, two main properties are necessary: first, models must learn from new incoming data — potentially from a different task -– with satisfactory asymptotic performance and sample complexity. This capability is called plasticity. Second, they must retain the knowledge from previously learned tasks, known as memory stability. When this does not happen, and the performance of previous tasks degrades, the model suffers from Catastrophic Forgetting [1, 2]. These two properties are the central core of Continual Learning (CL) [3, 4], being strongly relevant for ML systems susceptible to test-time distributional shifts.

Given the critical importance of this topic, extensive literature addresses the challenges of CL in traditional ML methods [3, 5, 2, 6] and, more

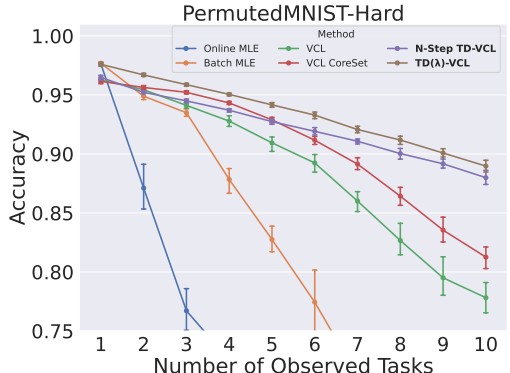

Figure 1: **Average accuracy across observed tasks in the PermutedMNIST-Hard benchmark**. The TD-VCL approach, proposed in this work, leads to a substantial improvement against standard VCL and non-variational approaches.

---

[*]Correspondence to: luckeciano.carvalho.melo@cs.ox.ac.uk
[†]Denotes equal supervision.

39th Conference on Neural Information Processing Systems (NeurIPS 2025).

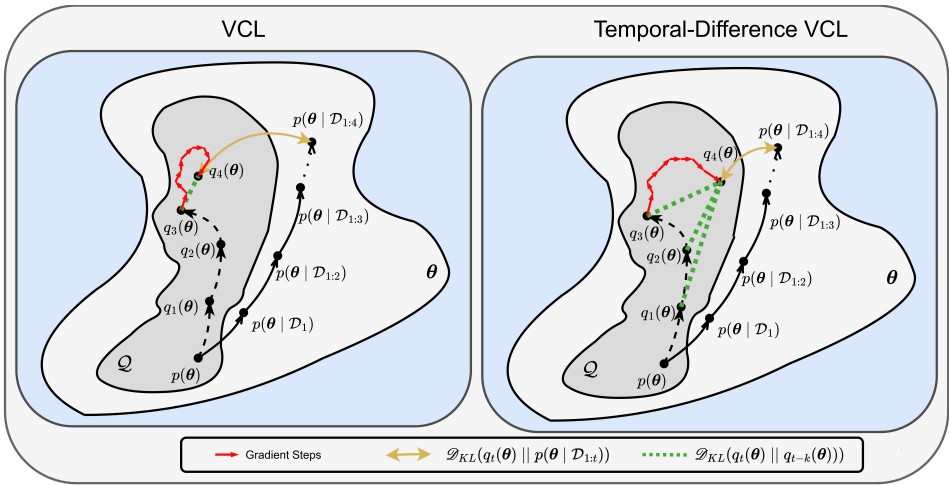

Figure 2: **An intuitive illustration of how TD-VCL functions in comparison to vanilla VCL**. At each timestep $t$, a new task dataset $\mathcal{D}_t$ arrives. Both methods aim to learn variational parameters $q_t(\boldsymbol{\theta})$ over a family of distributions $\mathcal{Q}$ that approximates the true posterior $p(\boldsymbol{\theta} \mid \mathcal{D}_{1:t})$ via minimizing the KL divergence $\mathcal{D}_{KL}(q_t(\boldsymbol{\theta}) \mid\mid p(\boldsymbol{\theta} \mid \mathcal{D}_{1:t}))$. VCL optimization (left) is only constrained by the most recent posterior, which compounds approximation errors from previous estimations and potentially deviates far from the true posterior. TD-VCL (right) is regularized by a sequence of past estimations, alleviating the impact of compounded errors.

recently, for overparameterized models [7, 1, 8]. In this work, we focus on Bayesian CL methods, for two reasons. First, it provides a principled, self-consistent framework for learning in online or low-data regimes [9]. Second, Bayesian models express their own uncertainty over predictions, which is crucial for safety-critical applications [10] and for enabling principled data selection [11, 12].

Particularly, we investigate Variational Continual Learning (VCL) approaches [13]. As detailed in Section 3, VCL identifies a recursive relationship between subsequent posterior distributions over tasks. A variational optimization objective then leverages this recursion, which regularizes the updated posterior to stay close to the very latest posterior approximation. Nevertheless, we argue that solely relying on a single previous posterior estimate for building up the next optimization target may be ineffective, as the approximation error propagates to the next update and compounds after successive recursions. If a particular estimation is especially poor, the error will be carried over to the next step entirely, which can dramatically degrade model's performance.

In this work, we show that the same optimization objective can be represented as a function of a sequence of previous posterior estimates and task likelihoods. We thus propose a new Continual Learning objective, n-Step KL VCL, that explicitly regularizes the posterior update considering several past posterior approximations. By considering multiple previous estimates, the objective dilutes individual errors, allows correct posterior approximates to exert a corrective influence, and leverages a broader global context to the learning target, reducing the impact of compounding errors over time. Figure 2 illustrates the underlying mechanism.

We further generalize this unbiased optimization target to a broader family of CL objectives, namely Temporal-Difference VCL, which constructs the learning target by prioritizing the most recent approximated posteriors. We reveal a link between the proposed objective and Temporal-Difference (TD) methods, a popular learning mechanism in Reinforcement Learning [14] and Neuroscience [15]. Furthermore, we show that TD-VCL represents a spectrum of learning objectives that range from vanilla VCL to n-Step KL VCL. Finally, we present experiments on several challenging and popular CL benchmarks, demonstrating that they outperform standard VCL (as shown in Figure 1), other VCL-based methods, and non-variational baselines, effectively alleviating Catastrophic Forgetting.

## 2 Related Work

**Continual Learning** has been studied throughout the past decades, both in Artificial Intelligence [3, 5, 16] and in Neuro- and Cognitive Sciences [17, 6, 2]. More recently, the focus has shifted

towards overparameterized models, such as deep neural networks [7, 1, 8, 18]. Given their powerful predictive capabilities, recent literature approaches CL from a wide range of perspectives. For instance, by regularizing the optimization objective to account for old tasks [19–21]; by replaying an external memory composed by a set of previous tasks [22–24]; or by modifying the optimization procedure or manipulating the estimated gradients [25–27]. We refer to Wang et al. for an extensive review of recent approaches. Our proposed method is placed between regularization-based and replay-based methods.

**Bayesian CL.** In the Bayesian framework, prior methods exploit the recursive relationship between subsequent posteriors that emerge from the Bayes' rule in the CL setting (Section 3). Since Bayesian inference is often intractable, they fundamentally differ in the design of approximated inference. We highlight works that learn posteriors via Laplace approximation [29, 30], sequential Bayesian Inference [31, 32], and Variational Inference (VI) [13, 33, 34]. Our work and proposed method lies in the latter category.

**Variational Inference for CL.** Variational Continual Learning (VCL) [13] introduced the idea of online VI for the Continual Learning setting. It leverages the Bayesian recursion of posteriors to build an optimization target for the next step's posterior based on the current one. Similarly, our work also optimizes a target based on previous approximated posteriors. On the other hand, rather than relying on a single past posterior estimation, it bootstraps on several previous estimations to prevent compounded errors. Nguyen et al. [13] further incorporate an heuristic external replay buffer to prevent forgetting, requiring a two-step optimization. In contrast, our work only requires a single-step optimization as the replay mechanism naturally emerges from the learning objective.

Other derivative works usually blend VCL with architectural and optimization improvements [35, 33, 36–40] or different posterior modeling assumptions [34, 41–44]. We specifically highlight UCB [38], which adapts the learning rate according to the uncertainty of the Bayesian model, and UCL [43], which introduces a different implementation for the VCL objective by proposing the notion of node-wise uncertainty. While their contribution are orthogonal to ours, we adopt UCB and UCL as comparison methods to further show that our proposed objective may also be combined with other variational methods and enhance their performance.

## 3 Preliminaries

**Problem Statement**. In the Continual Learning setting, a model learns from a streaming of tasks, which forms a non-stationary data distribution throughout time. More formally, we consider a task distribution $\mathcal{T}$ and represent each task $t \sim \mathcal{T}$ as a set of pairs $\{(\boldsymbol{x}_t, y_t)\}^{N_t}$, where $N_t$ is the dataset size. At every timestep $t^3$, the model receives a batch of data $\mathcal{D}_t$ for training. We evaluate the model in held-out test sets, considering all previously observed tasks.

In the **Bayesian framework** for CL, we assume a prior distribution over parameters $p(\boldsymbol{\theta})$, and the goal is to learn a posterior distribution $p(\boldsymbol{\theta} \mid \mathcal{D}_{1:T})$ after observing $T$ tasks. Crucially, given the sequential nature of tasks, we identify a recursive property of posteriors:

$$p(\boldsymbol{\theta} \mid \mathcal{D}_{1:T}) \propto p(\boldsymbol{\theta})p(\mathcal{D}_{1:T} \mid \boldsymbol{\theta}) \overset{\text{i.i.d}}{=} p(\boldsymbol{\theta}) \prod_{t=1}^{T} p(\mathcal{D}_t \mid \boldsymbol{\theta}) \propto p(\boldsymbol{\theta} \mid \mathcal{D}_{1:T-1})p(\mathcal{D}_T \mid \boldsymbol{\theta}), \qquad (1)$$

where we assume that tasks are i.i.d. Equation 1 shows that we may update the posterior estimation online, given the likelihood of the subsequent task.

**Variational Continual Learning**. Despite the elegant recursion, computing the posterior $p(\boldsymbol{\theta} \mid \mathcal{D}_{1:T})$ exactly is often intractable, especially for large parameter spaces. Hence, we rely on an approximation. VCL achieves this by employing online variational inference [45]. It assumes the existence of variational parameters $q(\boldsymbol{\theta})$ whose goal is to approximate the posterior by minimizing the following KL divergence over a space of variational approximations $\mathcal{Q}$:

$$q_t(\boldsymbol{\theta}) = \underset{q \in \mathcal{Q}}{\arg\min} \, \mathscr{D}_{KL}(q(\boldsymbol{\theta}) \mid\mid \frac{1}{Z_t} q_{t-1}(\boldsymbol{\theta})p(\mathcal{D}_t \mid \boldsymbol{\theta})), \qquad (2)$$

---

[3]For notational simplicity, we use the index $t$ to denote both tasks and timesteps. Note that neither the VCL framework nor our proposed methodology requires knowledge of task boundaries, as argued in the Appendix N.

where $Z_t$ represents a normalization constant. The objective in Equation 2 is equivalent to maximizing the variational lower bound of the online marginal likelihood:

$$\mathcal{L}_{VCL}^t(\boldsymbol{\theta}) = \mathbb{E}_{\boldsymbol{\theta} \sim q_t(\boldsymbol{\theta})}[\log p(\mathcal{D}_t \mid \boldsymbol{\theta})] - \mathscr{D}_{KL}(q_t(\boldsymbol{\theta}) \mid\mid q_{t-1}(\boldsymbol{\theta})). \tag{3}$$

We can interpret the loss in Equation 3 through the lens of the stability-plasticity dilemma [4]. The first term maximizes the likelihood of the new task (encouraging plasticity), whereas the KL term penalizes parametrizations that deviate too far from the previous posterior estimation, which supposedly contains the knowledge from past tasks (encouraging memory stability).

## 4 Temporal-Difference Variational Continual Learning

Maximizing the objective in Equation 3 is equivalent to the optimization in Equation 2, but its computation relies on two main approximations. First, computing the expected log-likelihood term analytically is not tractable, which requires a Monte-Carlo (MC) approximation. Second, the KL term relies on a previous posterior estimate, which may be biased from previous approximation errors. While updating the posterior to account for the next task, these biases deviate the learning target from the true objective. Crucially, as Equation 3 solely relies on the very latest posterior estimation, the error compounds with successive recursive updates.

Alternatively, we may represent the same objective as a function of several previous posterior estimations and alleviate the effect of the approximation error from any particular one. By considering several past estimates, the objective dilutes individual errors, allows correct posterior approximates to exert a corrective influence, and leverages a broader global context to the learning target, reducing the impact of compounding errors over time.

### 4.1 Variational Continual Learning with n-Step KL Regularization

We start by presenting a new objective that is equivalent to Equation 2 while also meeting the aforementioned desiderata:

**Proposition 4.1.** *The standard KL minimization objective in Variational Continual Learning (Equation 2) is equivalently represented as the following objective, where $n \in \mathbb{N}_0$ is a hyperparameter:*

$$q_t(\boldsymbol{\theta}) = \arg\max_{q \in \mathcal{Q}} \mathbb{E}_{\boldsymbol{\theta} \sim q_t(\boldsymbol{\theta})} \Big[ \sum_{i=0}^{n-1} \frac{(n-i)}{n} \log p(\mathcal{D}_{t-i} \mid \boldsymbol{\theta}) \Big] - \sum_{i=0}^{n-1} \frac{1}{n} \mathscr{D}_{KL}(q_t(\boldsymbol{\theta}) \mid\mid q_{t-i-1}(\boldsymbol{\theta})). \tag{4}$$

We present the proof of Proposition 4.1 in **Appendix A**. We name Equation 4 as the n-Step KL regularization objective. It represents the same learning target of Equation 2 as a sum of weighted likelihoods and KL terms that consider different posterior estimations, which can be interpreted as "distributing" the role of regularization among them. For instance, if an estimate $q_{t-i}$ deviates too far from the true posterior, it only affects $1/n$ of the KL regularization term. The hyperparameter $n$ assumes integer values up to $t$ and defines how far in the past the learning target goes. If $n$ is set to 1, we recover vanilla VCL.

An interesting insight comes from the likelihood term. It contains the likelihood of different tasks, weighted by their recency. Hence, the idea of re-estimating old task likelihoods, commonly leveraged as a heuristic in CL methods, fundamentally emerges in the proposed objective. We may estimate these likelihood terms by replaying data from different tasks simultaneously, alleviating the violation of the i.i.d assumption that happens given the online, sequential nature of CL [7].

### 4.2 From n-Step KL to Temporal-Difference Targets

The learning objective in Equation 4 relies on several different posterior estimates, alleviating the compounding error problem. A caveat is that all estimates have the same weight in the final objective. One may want to have more flexibility by giving different weights for them – for instance, amplifying the effect from the most recent estimate while drastically reducing the impact of previous ones. It is possible to accomplish that, as shown in the following proposition:

**Proposition 4.2.** *The standard KL minimization objective in VCL (Equation 2) is equivalently represented as the following objective, with $n \in \mathbb{N}_0$, and $\lambda \in [0,1)$ hyperparameters:*

$$q_t(\boldsymbol{\theta}) = \underset{q \in \mathcal{Q}}{\arg\max} \mathbb{E}_{\boldsymbol{\theta} \sim q_t(\boldsymbol{\theta})} \bigg[ \sum_{i=0}^{n-1} \frac{\lambda^i (1-\lambda^{n-i})}{1-\lambda^n} \log p(\mathcal{D}_{t-i} \mid \boldsymbol{\theta}) \bigg]$$
$$- \sum_{i=0}^{n-1} \frac{\lambda^i (1-\lambda)}{1-\lambda^n} \mathscr{D}_{KL}(q_t(\boldsymbol{\theta}) \;||\; q_{t-i-1}(\boldsymbol{\theta})). \tag{5}$$

The proof is available in **Appendix B**. We call Equation 5 the TD($\lambda$)-VCL objective[4]. It augments the n-Step KL Regularization to weight the regularization effect of different estimates in a way that geometrically decays – via the $\lambda^i$ term – as far as it goes in the past. Other $\lambda$-related terms serve as normalization constants. Equation 5 provides a more granular level of target control.

Interestingly, this objective relates intrinsically to the $\lambda$-returns for Temporal-Difference (TD) learning in valued-based reinforcement learning [46]. More broadly, both objectives of Equations 4 and 5 are compound updates that combine $n$-step Temporal-Difference targets, as shown below. First, we formally define a TD target in the CL context:

**Definition 4.3.** For a timestep $t$, the n-Step Temporal-Difference target for Variational Continual Learning is defined as, $\forall n \in \mathbb{N}_0$, $n \leq t$:

$$\text{TD}_t(n) = \mathbb{E}_{\boldsymbol{\theta} \sim q_t(\boldsymbol{\theta})} \bigg[ \sum_{i=0}^{n-1} \log p(\mathcal{D}_{t-i} \mid \boldsymbol{\theta}) \bigg] - \mathscr{D}_{KL}(q_t(\boldsymbol{\theta}) \;||\; q_{t-n}(\boldsymbol{\theta})). \tag{6}$$

In **Appendix C**, we reveal the connection between Equation 6 and the TD targets employed in Reinforcement Learning, justifying the adopted terminology. From this definition, it follows that:

**Proposition 4.4.** $\forall n \in \mathbb{N}_0$, $n \leq t$, *the objective in Equation 2 can be equivalently represented as:*

$$q_t(\boldsymbol{\theta}) = \underset{q \in \mathcal{Q}}{\arg\max} \, \text{TD}_t(n), \tag{7}$$

*with* $\text{TD}_t(n)$ *as in Definition 4.3. Furthermore, the objective in Equation 5 can also be represented as:*

$$q_t(\boldsymbol{\theta}) = \underset{q \in \mathcal{Q}}{\arg\max} \frac{1-\lambda}{1-\lambda^n} \underbrace{\bigg[ \sum_{k=0}^{n-1} \lambda^k \text{TD}_t(k+1) \bigg]}_{\text{Discounted sum of TD targets}}. \tag{8}$$

The proof is in **Appendix D**. Proposition 4.4 states that the TD($\lambda$)-VCL objective is a sum of discounted TD targets (up to a normalization constant), effectively representing $\lambda$-returns. In parallel, one can show that the n-Step KL Regularization objective, as a particular case, is a simple average of n-Step TD targets. Fundamentally, the key idea behind these objectives is *bootstrapping*: they build a learning target estimate based on other estimates. Ultimately, the "$\lambda$-target" in Equation 5 provides flexibility for bootstrapping by allowing multiple previous estimates to influence the objective.

**The TD-VCL objectives generalize a spectrum of Continual Learning algorithms**. As a final remark, in **Appendix E**, we show that, based on the choice of hyperparameters, the TD($\lambda$)-VCL objective forms a family of learning algorithms that span from Vanilla VCL to n-Step KL Regularization. Fundamentally, it mixes different targets of MC approximations for expected log-likelihood and KL regularization. This process is similar to how TD($\lambda$) and $n$-step TD mix MC updates and TD predictions in Reinforcement Learning, effectively providing a mechanism to strike a balance between the variance from MC estimations and the bias from bootstrapping [46].

---

[4]We refer to both n-Step KL Regularization and TD($\lambda$)-VCL as TD-VCL objectives.

# 5 Experiments and Discussion

Our central hypothesis is that for Bayesian CL, leveraging multiple past posterior estimates mitigates the impact of compounded errors inherent to the VCL objective, thus alleviating the problem of Catastrophic Forgetting. We now provide an experimental setup for validation. Specifically, we evaluate this hypothesis by analyzing the questions highlighted in Section 5.1.

**Implementation**. We use a Gaussian mean-field approximate posterior and assume a Gaussian prior $\mathcal{N}(0, \sigma^2 I)$, and parameterize all distributions as deep networks. For all variational objectives, we compute the KL term analytically and employ Monte Carlo approximations for the expected log-likelihood terms, leveraging the reparametrization trick [47] for computing gradients. We employed likelihood-tempering [33] to prevent variational over-pruning [48]. Lastly, for test-time evaluation, we compute the posterior predictive distribution by marginalizing out the approximated posterior via Monte-Carlo sampling. We provide further detail about architecture and training in Appendix G and our code[5].

**Comparison Methods**. We compare TD-VCL and n-Step KL VCL against several methods. We first evaluate non-variational naive methods for CL: **Online MLE** naively applies maximum likelihood estimation in the current task data. It serves as a lower bound for other methods, as well as a way to evaluate how challenging the benchmark is. **Batch MLE** applies maximum likelihood estimation considering a buffer of current and old task data. Next, we adopt the following variational methods for direct comparison in the Bayesian CL setting: **VCL**, introduced by Nguyen et al. [13], optimizes the objective in Equation 3. **VCL CoreSet** is a VCL variant that incorporates a replay set to mitigate any residual forgetting [13]. **UCL** [43] is another variational method that implements adaptive regularization based on the notion of node-wise uncertainty. Finally, **UCB** [38] also optimizes the objective of Equation 3 but adapts the learning rate for each parameter based on their uncertainty. Particularly for UCL and UCB, we compare them with the proposed **TD-UCL** and **TD-UCB**, which incorporate the introduced objective into UCL and UCB, respectively.

**Benchmarks**. We evaluate five benchmarks for Continual Learning (CL). First, we introduce three new benchmarks: **PermutedMNIST-Hard**, **SplitMNIST-Hard**, and **SplitNotMNIST-Hard**. These are more challenging versions of traditional CL benchmarks with similar names. They are significantly harder due to two key restrictions. First, the amount of replay memory that any method can use is limited in both dataset size and the number of tasks. As empirically shown in Appendix I, this creates a much more acute scenario of Catastrophic Forgetting. Second, they enforce the adoption of single-head classifiers. As also shown in Appendix H, this requires the model to account for the potential negative transfer learning among tasks, which makes MNIST/NotMNIST-based benchmarks non-trivial for current research. Next, we also evaluate on two other popular CL benchmarks: **CIFAR100-10** and **TinyImageNet-10**. Both benchmarks are very challenging classification problems, particularly in our setting where no pre-trained representations are used. In Appendix J, we detail all benchmark tasks and specific constraints adopted for robust evaluation.

## 5.1 Experiments

We highlight and analyze the following questions to evaluate our hypothesis and proposed method:

**Do the TD-VCL objectives effectively alleviate Catastrophic Forgetting in challenging CL benchmarks?** Tables 1 and 2 present the results for all benchmarks. Each column presents the average accuracy across the past $t$ observed tasks, and we show the results starting from $t = 2$ as $t = 1$ is simply single-task learning. For **PermutedMNIST-Hard**, all methods present high accuracy for $t = 2$, suggesting that they could fit the data successfully. As the number of tasks increases, they start manifesting Catastrophic Forgetting at different levels. While Online and Batch MLE drastically suffer, variational approaches considerably retain old tasks' performance. The Core Set slightly helps VCL, and both n-Step KL and TD-VCL outperform them by a considerable margin, attaining approximately 90% average accuracy after all tasks. For completeness, Figure 1 graphically shows the results. We emphasize the discrepancy between variational approaches and naive baselines and highlight the performance boost by adopting TD-VCL objectives.

For **SplitMNIST-Hard**, we highlight that the TD-VCL objectives also surpass baselines in all configurations, but with a decrease in performance for $t = 5$, suggesting a more challenging setup for

---

[5]Our code is available at `https://github.com/luckeciano/TD-VCL`

Table 1: **Quantitative comparison on the PermutedMNIST-Hard, SplitMNIST-Hard, and SplitNotMNIST-Hard benchmarks**. Each column presents the average accuracy across the past $t$ observed tasks. Results are reported with two standard deviations across ten seeds. Top two results are in **bold**, while noticeably lower results are in gray. TD-VCL objective consistently outperforms standard VCL variants, especially when the number of observed tasks increase.

| | PermutedMNIST-Hard | | | | | | | | |
| | t = 2 | t = 3 | t = 4 | t = 5 | t = 6 | t = 7 | t = 8 | t = 9 | t = 10 |
| --- | --- | --- | --- | --- | --- | --- | --- | --- | --- |
| Online MLE | 0.87±0.07 | 0.77±0.06 | 0.73±0.08 | 0.69±0.08 | 0.65±0.13 | 0.57±0.16 | 0.51±0.14 | 0.46±0.11 | 0.40±0.08 |
| Batch MLE | 0.95±0.01 | 0.93±0.01 | 0.88±0.04 | 0.83±0.04 | 0.77±0.10 | 0.71±0.13 | 0.64±0.12 | 0.57±0.11 | 0.51±0.06 |
| VCL | 0.95±0.00 | 0.94±0.01 | 0.93±0.02 | 0.91±0.02 | 0.89±0.03 | 0.86±0.03 | 0.83±0.04 | 0.80±0.06 | 0.78±0.04 |
| VCL CoreSet | **0.96±0.00** | **0.95±0.00** | **0.94±0.00** | **0.93±0.02** | 0.91±0.01 | 0.89±0.02 | 0.86±0.03 | 0.84±0.04 | 0.81±0.03 |
| n-Step TD-VCL | 0.95±0.01 | 0.94±0.00 | **0.94±0.00** | **0.93±0.01** | **0.92±0.01** | **0.91±0.01** | **0.90±0.02** | **0.89±0.01** | **0.88±0.02** |
| TD($\lambda$)-VCL | **0.97±0.00** | **0.96±0.00** | **0.95±0.00** | **0.94±0.01** | **0.93±0.01** | **0.92±0.01** | **0.91±0.01** | **0.90±0.01** | **0.89±0.02** |

| | SplitMNIST-Hard | | | | | SplitNotMNIST-Hard | | | |
| | t = 2 | t = 3 | t = 4 | t = 5 | | t = 2 | t = 3 | t = 4 | t = 5 |
| --- | --- | --- | --- | --- | --- | --- | --- | --- | --- |
| Online MLE | 0.86±0.02 | 0.61±0.03 | 0.75±0.04 | 0.57±0.06 | | 0.72±0.02 | 0.61±0.05 | 0.61±0.00 | 0.51±0.04 |
| Batch MLE | 0.95±0.04 | 0.65±0.04 | 0.82±0.04 | 0.59±0.03 | | 0.71±0.02 | 0.65±0.03 | 0.61±0.00 | 0.50±0.06 |
| VCL | 0.87±0.02 | 0.66±0.04 | 0.82±0.03 | 0.64±0.11 | | 0.69±0.04 | 0.63±0.03 | 0.60±0.00 | 0.51±0.06 |
| VCL CoreSet | 0.93±0.04 | 0.68±0.07 | 0.84±0.04 | 0.62±0.03 | | 0.69±0.04 | 0.65±0.02 | 0.60±0.01 | 0.51±0.07 |
| n-Step TD-VCL | **0.98±0.01** | **0.79±0.08** | **0.88±0.04** | **0.67±0.04** | | **0.72±0.04** | **0.73±0.05** | **0.70±0.04** | **0.58±0.08** |
| TD($\lambda$)-VCL | **0.98±0.01** | **0.81±0.07** | **0.89±0.03** | **0.66±0.02** | | **0.74±0.02** | **0.73±0.03** | **0.69±0.03** | **0.58±0.09** |

Table 2: **Quantitative comparison on the CIFAR100-10 and TinyImagenet-10 benchmarks**. Each column presents the average accuracy across the past $t$ observed tasks. Results are reported with two standard deviations across five seeds. TD-VCL variants consistently outperform the baselines in harder benchmarks with more complex architectures, such as Bayesian CNNs. The full table is available in the Appendix M.

| | CIFAR100-10 | | | | TinyImageNet-10 | | | |
| | t = 4 | t = 6 | t = 8 | t = 10 | t = 4 | t = 6 | t = 8 | t = 10 |
| --- | --- | --- | --- | --- | --- | --- | --- | --- |
| Online MLE | 0.57±0.06 | 0.56±0.03 | 0.53±0.06 | 0.52±0.04 | 0.45±0.02 | 0.44±0.01 | 0.45±0.02 | 0.44±0.03 |
| Batch MLE | 0.58±0.04 | 0.58±0.05 | 0.56±0.06 | 0.54±0.07 | 0.48±0.02 | 0.48±0.02 | 0.50±0.02 | 0.51±0.03 |
| VCL | 0.63±0.02 | 0.60±0.02 | 0.61±0.05 | 0.66±0.01 | 0.51±0.03 | 0.51±0.03 | 0.51±0.02 | 0.51±0.02 |
| VCL CoreSet | 0.63±0.03 | 0.63±0.02 | 0.61±0.02 | 0.65±0.02 | 0.51±0.02 | 0.51±0.02 | 0.54±0.02 | 0.54±0.02 |
| n-Step TD-VCL | **0.67±0.02** | **0.65±0.01** | **0.68±0.04** | **0.69±0.02** | **0.55±0.02** | 0.54±0.02 | **0.56±0.02** | **0.56±0.02** |
| TD($\lambda$)-VCL | **0.66±0.04** | **0.66±0.02** | **0.67±0.01** | **0.71±0.01** | **0.56±0.02** | **0.55±0.03** | **0.56±0.02** | **0.56±0.02** |

addressing Catastrophic Forgetting that opens a venue for future research. We discuss SplitMNIST-Hard results in more detail in Appendix K. Next, **SplitNotMNIST-Hard** is a harder benchmark, as the letters come from a diverse set of font styles. Furthermore, we purposely decided to employ a modest network architecture (as for previous benchmarks). Facing hard tasks with less expressive parametrizations will result in higher posterior approximation error. Our goal is to evaluate how the variational methods behave in this setting. Once again, n-step KL and TD-VCL surpassed the baselines after observing more than three tasks. The effect is more pronounced after increasing the number of observed tasks. These objectives are the only ones whose resultant models achieved non-trivial average accuracy after observing all tasks.

Lastly, we analyze the results on **CIFAR100-10** and **TinyImageNet-10** in Table 2. These are considerably harder benchmarks, as the distribution of images and classes is much richer than the previous benchmarks. Furthermore, they necessarily require better architectures to attain non-trivial performance. Following previous work [8, 49, 50], we adopt an AlexNet architecture [51]. This setup is ideal for evaluating how the learning objective functions at a larger scale with more complex, deep architectures such as (Bayesian) convolutional networks. Once again, TD-VCL objectives attain superior performance, particularly for later timesteps, where Catastrophic Forgetting is more pronounced in the baselines. This suggests that leveraging multiple posterior estimates for learning is better than only the latest one, even when the approximation error is high.

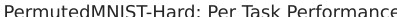

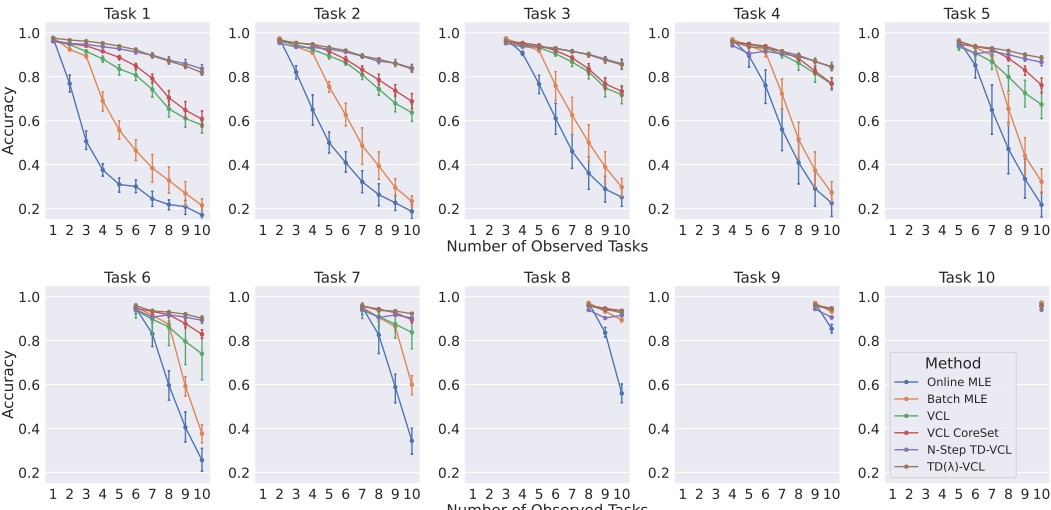

Figure 3: **Per-task performance (accuracy) over time in the PermutedMNIST-Hard benchmark**. Each plot represents the accuracy of one task (identified in the plot title) while the number of observed tasks increases. We highlight a stronger effect of Catastrophic Forgetting on earlier tasks for the baselines, while TD-VCL objectives are noticeably more robust to this phenomenon.

**How do the TD-VCL objectives affect per-task performance?** While the previous question analyze the performance averaged across different tasks, we now investigate the accuracy of each task separately in the course of online learning. This setup is relevant since solely considering the averaged accuracy may hide a stronger Catastrophic Forgetting effect from earlier tasks by "compensating" with higher accuracy from later tasks. We show the results for PermutedMNIST-Hard in Figure 3 (we defer additional per-task results for Appendix K). It presents a sequence of plots, where each figure represents the accuracy of one task while the number of observed tasks increases. Naturally, the tasks that appear at later stages present fewer data points: for instance, "Task 10" has a single data point as it does not have test data for earlier timesteps.

As observed, per-task performance explicitly shows a stronger effect of Catastrophic Forgetting for earlier tasks in the adopted baselines. We particularly highlight how non-variational approaches fail for them. In this direction, TD-VCL objectives presented a more robust performance against others. For instance, we highlight the results for Task 1. After observing all tasks, the proposed methods demonstrated accuracy of around 80% and 85%. The VCL baselines dropped to 50% and 60%, and MLE-based methods failed with only 20% of accuracy.

**How does TD-VCL (and variants) perform against other Bayesian CL methods?**

In this work, we focus on Continual Learning with a Bayesian lens. As highlighted in Section 1, it provides a formal, uncertainty-aware framework crucial for safety-critical applications and data-efficient learning. Thus, we analyze the TD objective (Equation 5) on other Bayesian CL methods. UCL and UCB are variational methods that optimize the objective in Equation 2 but propose new mechanisms for regularization and learning rate adaptation. Since these enhancements are orthogonal to the objective, we incorporate the proposed TD objective with these methods, resulting in TD-UCL and TD-UCB, respectively. We aim to show that the TD objectives for CL work across different base methods and promote a performance boost on them.

Table 3 compares the base methods (VCL, UCL, and UCB) with their TD-enhanced counterparts (complete results in Appendix M). While there is no dominant base method across the benchmarks, the TD counterparts consistently improve upon their respective base methods, especially at later timesteps. These results indicate that the TD objective is robust among different Bayesian CL algorithms and may be incorporated effectively into methods that rely on the variational objective in Equation 2.

**How do the TD-VCL objectives behave with the choice of the hyperparameters $n$, $\lambda$, and the likelihood-tempering parameter $\beta$?** The proposed learning objectives introduce two new

Table 3: **Quantitative comparison between Bayesian CL methods and their TD-enhanced counterparts**. The TD-enhanced methods incorporate the objective in Equation 5 in each base method. Although no single base method consistently outperforms the others across all benchmarks, their TD-enhanced versions consistently achieve better performance, particularly at later timesteps. The full table is avaliable in Appendix M.

| | PermutedMNIST-Hard | | | | SplitMNIST-Hard | | | |
|---|---|---|---|---|---|---|---|---|
| | $t = 4$ | $t = 6$ | $t = 8$ | $t = 10$ | $t = 2$ | $t = 3$ | $t = 4$ | $t = 5$ |
| VCL | $0.93_{\pm0.02}$ | $0.89_{\pm0.03}$ | $0.83_{\pm0.04}$ | $0.78_{\pm0.04}$ | $0.87_{\pm0.02}$ | $0.66_{\pm0.04}$ | $0.82_{\pm0.03}$ | $0.64_{\pm0.11}$ |
| TD($\lambda$)-VCL | $\mathbf{0.95_{\pm0.00}}$ | $\mathbf{0.93_{\pm0.01}}$ | $\mathbf{0.91_{\pm0.01}}$ | $\mathbf{0.89_{\pm0.02}}$ | $\mathbf{0.98_{\pm0.01}}$ | $\mathbf{0.79_{\pm0.08}}$ | $\mathbf{0.88_{\pm0.04}}$ | $\mathbf{0.67_{\pm0.04}}$ |
| UCL | $0.94_{\pm0.00}$ | $0.89_{\pm0.02}$ | $0.83_{\pm0.06}$ | $0.73_{\pm0.12}$ | $0.88_{\pm0.04}$ | $0.68_{\pm0.03}$ | $0.83_{\pm0.03}$ | $0.66_{\pm0.06}$ |
| TD($\lambda$)-UCL | $\mathbf{0.95_{\pm0.00}}$ | $\mathbf{0.92_{\pm0.02}}$ | $\mathbf{0.88_{\pm0.04}}$ | $\mathbf{0.84_{\pm0.04}}$ | $\mathbf{0.97_{\pm0.01}}$ | $\mathbf{0.85_{\pm0.06}}$ | $\mathbf{0.90_{\pm0.02}}$ | $\mathbf{0.70_{\pm0.04}}$ |
| UCB | $0.92_{\pm0.01}$ | $0.89_{\pm0.02}$ | $0.86_{\pm0.02}$ | $0.83_{\pm0.02}$ | $0.85_{\pm0.16}$ | $0.79_{\pm0.12}$ | $0.83_{\pm0.06}$ | $0.75_{\pm0.10}$ |
| TD($\lambda$)-UCB | $\mathbf{0.93_{\pm0.00}}$ | $\mathbf{0.91_{\pm0.01}}$ | $\mathbf{0.90_{\pm0.01}}$ | $\mathbf{0.88_{\pm0.02}}$ | $\mathbf{0.93_{\pm0.02}}$ | $\mathbf{0.89_{\pm0.03}}$ | $\mathbf{0.87_{\pm0.03}}$ | $\mathbf{0.80_{\pm0.03}}$ |

| | CIFAR100-10 | | | | TinyImageNet-10 | | | |
|---|---|---|---|---|---|---|---|---|
| | $t = 4$ | $t = 6$ | $t = 8$ | $t = 10$ | $t = 4$ | $t = 6$ | $t = 8$ | $t = 10$ |
| VCL | $0.63_{\pm0.02}$ | $0.60_{\pm0.02}$ | $0.61_{\pm0.05}$ | $0.66_{\pm0.01}$ | $0.51_{\pm0.03}$ | $0.51_{\pm0.03}$ | $0.51_{\pm0.02}$ | $0.51_{\pm0.02}$ |
| TD($\lambda$)-VCL | $\mathbf{0.66_{\pm0.04}}$ | $\mathbf{0.66_{\pm0.02}}$ | $\mathbf{0.67_{\pm0.01}}$ | $\mathbf{0.71_{\pm0.01}}$ | $\mathbf{0.56_{\pm0.02}}$ | $\mathbf{0.55_{\pm0.03}}$ | $\mathbf{0.56_{\pm0.02}}$ | $\mathbf{0.56_{\pm0.06}}$ |
| UCL | $0.64_{\pm0.05}$ | $0.60_{\pm0.05}$ | $0.58_{\pm0.02}$ | $0.62_{\pm0.02}$ | $0.52_{\pm0.03}$ | $0.51_{\pm0.02}$ | $0.52_{\pm0.02}$ | $0.50_{\pm0.03}$ |
| TD($\lambda$)-UCL | $\mathbf{0.64_{\pm0.01}}$ | $\mathbf{0.70_{\pm0.02}}$ | $\mathbf{0.66_{\pm0.03}}$ | $\mathbf{0.67_{\pm0.03}}$ | $\mathbf{0.54_{\pm0.01}}$ | $\mathbf{0.54_{\pm0.01}}$ | $\mathbf{0.55_{\pm0.01}}$ | $\mathbf{0.56_{\pm0.01}}$ |
| UCB | $0.66_{\pm0.02}$ | $0.66_{\pm0.03}$ | $0.65_{\pm0.01}$ | $0.66_{\pm0.01}$ | $0.51_{\pm0.02}$ | $0.48_{\pm0.04}$ | $0.45_{\pm0.02}$ | $0.42_{\pm0.03}$ |
| TD($\lambda$)-UCB | $\mathbf{0.66_{\pm0.01}}$ | $\mathbf{0.67_{\pm0.01}}$ | $\mathbf{0.68_{\pm0.01}}$ | $\mathbf{0.70_{\pm0.01}}$ | $\mathbf{0.52_{\pm0.01}}$ | $\mathbf{0.51_{\pm0.02}}$ | $\mathbf{0.50_{\pm0.03}}$ | $\mathbf{0.47_{\pm0.02}}$ |

hyperparameters: $n$ (the number of considered previous posterior estimates in the learning target) and $\lambda$ for TD($\lambda$)-VCL (which controls the level of influence for each past posterior estimate). Furthermore, it also inherits the $\beta$ parameter from VCL. Hence, we evaluate the sensitivity of the proposed objectives concerning these hyperparameters, presenting results and detailed discussion in Appendix L. We highlight three main findings. First, similarly to VCL, TD-VCL objectives are sensitive to the likelihood-tempering hyperparameter. Second, increasing $n$ is beneficial up to a certain point, from which it becomes detrimental, suggesting the existence of an optimal range for leveraging posterior estimates. Lastly, TD-VCL objectives present robustness over the choice of $\lambda$, with a more pronounced effect when the number of observed tasks increases.

## 6 Closing Remarks

In this work, we presented a new family of variational objectives for Continual Learning, namely Temporal-Difference VCL. TD-VCL is an unbiased proxy of the standard VCL objective but leverages several previous posterior estimates to alleviate the compounding error caused by recursive approximations. We showed that TD-VCL represents a spectrum of Continual Learning algorithms and is equivalent to a discounted sum of n-step Temporal-Difference targets. Lastly, we empirically presented that it helps address Catastrophic Forgetting, surpassing Bayesian CL baselines in several challenging benchmarks.

**Limitations**. Despite being theoretically principled and attaining superior performance, TD-VCL presents limitations. First, the hyperparameters $n$ and $\lambda$ depend on the evaluated setting, which may require certain tuning. Second, the objectives rely on past posterior estimates, which may increase memory requirements. Still, we believe this is not a major limitation as TD-VCL suits well modern deep Bayesian architectures that target smaller parameter subspaces for posterior approximation [52, 53, 12].

**Future Work**. While presenting connections with Temporal-Difference methods, TD-VCL is not an RL algorithm. Further mathematical connections with Markov Decision/Reward Processes formalism are left as future work. Another interesting direction is to apply TD-VCL objectives for other problems that involve sequential variational inference, such as probabilistic meta-learning [54, 55].

## Acknowledgments and Disclosure of Funding

The authors thank Panagiotis Tigas for insightful discussions on variational inference. Luckeciano C. Melo acknowledges funding from the Air Force Office of Scientific Research (AFOSR) European Office of Aerospace Research & Development (EOARD) under grant number FA8655-21-1-7017. Yarin Gal is supported by a Turing AI Fellowship financed by the UK government's Office for Artificial Intelligence, through UK Research and Innovation (grant reference EP/V030302/1) and delivered by the Alan Turing Institute.

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

# A  Derivation of the n-Step KL Regularization Objective

In this Section, we prove Proposition 4.1:

**Proposition 4.1.** *The standard KL minimization objective in Variational Continual Learning (Equation 2) is equivalently represented as the following objective, where $n \in \mathbb{N}_0$ is a hyperparameter:*

$$q_t(\boldsymbol{\theta}) = \arg\max_{q \in \mathcal{Q}} \mathbb{E}_{\boldsymbol{\theta} \sim q_t(\boldsymbol{\theta})} \Big[ \sum_{i=0}^{n-1} \frac{(n-i)}{n} \log p(\mathcal{D}_{t-i} \mid \boldsymbol{\theta}) \Big] - \sum_{i=0}^{n-1} \frac{1}{n} \mathscr{D}_{KL}(q_t(\boldsymbol{\theta}) \mid\mid q_{t-i-1}(\boldsymbol{\theta})). \quad (4)$$

*Proof.* Starting from Equation 2, we can expand it as a sum of equal terms and utilize the recursive property (Equation 1) to expand these terms:

$$
\begin{aligned}
q_t(\boldsymbol{\theta}) &= \arg\min_{q \in \mathcal{Q}} \mathscr{D}_{KL}(q(\boldsymbol{\theta}) \mid\mid \frac{1}{Z_t} q_{t-1}(\boldsymbol{\theta}) p(\mathcal{D}_t \mid \boldsymbol{\theta})) \\
&= \arg\min_{q \in \mathcal{Q}} \frac{n}{n} \mathscr{D}_{KL}(q(\boldsymbol{\theta}) \mid\mid \frac{1}{Z_t} q_{t-1}(\boldsymbol{\theta}) p(\mathcal{D}_t \mid \boldsymbol{\theta})) \\
&= \arg\min_{q \in \mathcal{Q}} \frac{1}{n} \Bigg[ \mathscr{D}_{KL}(q(\boldsymbol{\theta}) \mid\mid \frac{1}{Z_t} q_{t-1}(\boldsymbol{\theta}) p(\mathcal{D}_t \mid \boldsymbol{\theta})) \\
&\qquad\qquad + \mathscr{D}_{KL}(q(\boldsymbol{\theta}) \mid\mid \frac{1}{Z_t Z_{t-1}} q_{t-2}(\boldsymbol{\theta}) p(\mathcal{D}_t \mid \boldsymbol{\theta}) p(\mathcal{D}_{t-1} \mid \boldsymbol{\theta})) + \dots \\
&\qquad\qquad + \mathscr{D}_{KL}(q(\boldsymbol{\theta}) \mid\mid \frac{1}{\prod_{i=0}^{n-1} Z_{t-i}} q_{t-n}(\boldsymbol{\theta}) \prod_{i=0}^{n-1} p(\mathcal{D}_{t-i} \mid \boldsymbol{\theta})) \Bigg] \\
&= \arg\min_{q \in \mathcal{Q}} \frac{1}{n} \Bigg[ \mathscr{D}_{KL}(q_t(\boldsymbol{\theta}) \mid\mid q_{t-1}(\boldsymbol{\theta})) - \mathbb{E}_{\boldsymbol{\theta} \sim q_t(\boldsymbol{\theta})}[\log p(\mathcal{D}_t \mid \boldsymbol{\theta})] \\
&\qquad\qquad + \mathscr{D}_{KL}(q_t(\boldsymbol{\theta}) \mid\mid q_{t-2}(\boldsymbol{\theta})) - \mathbb{E}_{\boldsymbol{\theta} \sim q_t(\boldsymbol{\theta})}[\log p(\mathcal{D}_t \mid \boldsymbol{\theta}) + \log p(\mathcal{D}_{t-1} \mid \boldsymbol{\theta})] + \dots \\
&\qquad\qquad + \mathscr{D}_{KL}(q_t(\boldsymbol{\theta}) \mid\mid q_{t-n}(\boldsymbol{\theta})) - \mathbb{E}_{\boldsymbol{\theta} \sim q_t(\boldsymbol{\theta})}[\sum_{i=0}^{n-1} \log p(\mathcal{D}_{t-i} \mid \boldsymbol{\theta})] \Bigg] \\
&= \arg\min_{q \in \mathcal{Q}} \frac{1}{n} \Bigg[ \sum_{i=0}^{n-1} \mathscr{D}_{KL}(q_t(\boldsymbol{\theta}) \mid\mid q_{t-i}(\boldsymbol{\theta})) - \mathbb{E}_{\boldsymbol{\theta} \sim q_t(\boldsymbol{\theta})} \Big[ n \log p(\mathcal{D}_t \mid \boldsymbol{\theta}) \\
&\qquad\qquad + (n-1) \log p(\mathcal{D}_{t-1} \mid \boldsymbol{\theta}) + \dots + \log p(\mathcal{D}_{t-n+1} \mid \boldsymbol{\theta}) \Big] \Bigg] \\
&= \arg\max_{q \in \mathcal{Q}} \mathbb{E}_{\boldsymbol{\theta} \sim q_t(\boldsymbol{\theta})} \Big[ \sum_{i=0}^{n-1} \frac{(n-i)}{n} \log p(\mathcal{D}_{t-i} \mid \boldsymbol{\theta}) \Big] - \sum_{i=0}^{n-1} \frac{1}{n} \mathscr{D}_{KL}(q_t(\boldsymbol{\theta}) \mid\mid q_{t-i-1}(\boldsymbol{\theta})).
\end{aligned}
$$

$$(9)$$

$\square$

# B   Derivation of the Temporal-Difference VCL Objective

Before proving Proposition 4.2, we start by presenting a well known result for the sum of geometric series:

**Lemma B.1.** *The finite sum of a geometric series with $n$ terms, common ratio $\lambda$ and initial term $a$ is given by:*

$$\sum_{k=0}^{n-1} \lambda^k a = \frac{a(1 - \lambda^n)}{(1 - \lambda)} \tag{10}$$

*Proof.* Let $s_n = \sum_{k=0}^{n} \lambda^k a$. Hence,

$$
\begin{aligned}
s_n - \lambda s_n &= \sum_{k=0}^{n-1} \lambda^k a - \lambda \sum_{k=0}^{n-1} \lambda^k a = a - a\lambda^n \\
&\iff s_n(1 - \lambda) = a(1 - \lambda^n) \\
&\iff s_n = \frac{a(1 - \lambda^n)}{(1 - \lambda)}.
\end{aligned}
\tag{11}
$$

$\square$

Now, we prove Proposition 4.2.

**Proposition 4.2.** *The standard KL minimization objective in VCL (Equation 2) is equivalently represented as the following objective, with $n \in \mathbb{N}_0$, and $\lambda \in [0, 1)$ hyperparameters:*

$$
q_t(\boldsymbol{\theta}) = \arg\max_{q \in \mathcal{Q}} \mathbb{E}_{\boldsymbol{\theta} \sim q_t(\boldsymbol{\theta})} \Big[ \sum_{i=0}^{n-1} \frac{\lambda^i(1 - \lambda^{n-i})}{1 - \lambda^n} \log p(\mathcal{D}_{t-i} \mid \boldsymbol{\theta}) \Big]
$$
$$
- \sum_{i=0}^{n-1} \frac{\lambda^i(1 - \lambda)}{1 - \lambda^n} \mathscr{D}_{KL}(q_t(\boldsymbol{\theta}) \parallel q_{t-i-1}(\boldsymbol{\theta})). \tag{5}
$$

*Proof.* We can use Lemma B.1 to expand the sum of KL terms:

$$q_t(\boldsymbol{\theta}) = \underset{q \in \mathcal{Q}}{\arg\min} \ \mathscr{D}_{KL}(q(\boldsymbol{\theta}) \ || \ \frac{1}{Z_t} q_{t-1}(\boldsymbol{\theta}) p(\mathcal{D}_t \mid \boldsymbol{\theta}))$$

$$= \underset{q \in \mathcal{Q}}{\arg\min} \ \frac{1-\lambda}{1-\lambda^n} \frac{1-\lambda^n}{1-\lambda} \mathscr{D}_{KL}(q(\boldsymbol{\theta}) \ || \ \frac{1}{Z_t} q_{t-1}(\boldsymbol{\theta}) p(\mathcal{D}_t \mid \boldsymbol{\theta}))$$

$$= \underset{q \in \mathcal{Q}}{\arg\min} \ \frac{1-\lambda}{1-\lambda^n} \Bigg[ \mathscr{D}_{KL}(q(\boldsymbol{\theta}) \ || \ \frac{1}{Z_t} q_{t-1}(\boldsymbol{\theta}) p(\mathcal{D}_t \mid \boldsymbol{\theta}))$$

$$+ \lambda \mathscr{D}_{KL}(q(\boldsymbol{\theta}) \ || \ \frac{1}{Z_t Z_{t-1}} q_{t-2}(\boldsymbol{\theta}) p(\mathcal{D}_t \mid \boldsymbol{\theta}) p(\mathcal{D}_{t-1} \mid \boldsymbol{\theta})) + \dots$$

$$+ \lambda^{n-1} \mathscr{D}_{KL}(q(\boldsymbol{\theta}) \ || \ \frac{1}{\prod_{i=0}^{n-1} Z_{t-i}} q_{t-i}(\boldsymbol{\theta}) \prod_{i=0}^{n-1} p(\mathcal{D}_{t-i} \mid \boldsymbol{\theta})) \Bigg]$$

$$= \underset{q \in \mathcal{Q}}{\arg\min} \ \frac{1-\lambda}{1-\lambda^n} \Bigg[ \mathscr{D}_{KL}(q_t(\boldsymbol{\theta}) \ || \ q_{t-1}(\boldsymbol{\theta})) - \mathbb{E}_{\boldsymbol{\theta} \sim q_t(\boldsymbol{\theta})}[\log p(\mathcal{D}_t \mid \boldsymbol{\theta})]$$

$$+ \lambda \mathscr{D}_{KL}(q_t(\boldsymbol{\theta}) \ || \ q_{t-2}(\boldsymbol{\theta})) - \lambda \mathbb{E}_{\boldsymbol{\theta} \sim q_t(\boldsymbol{\theta})}[\log p(\mathcal{D}_t \mid \boldsymbol{\theta}) + \log p(\mathcal{D}_{t-1} \mid \boldsymbol{\theta})] + \dots$$

$$+ \lambda^{n-1} \mathscr{D}_{KL}(q_t(\boldsymbol{\theta}) \ || \ q_{t-n}(\boldsymbol{\theta})) - \lambda^{n-1} \mathbb{E}_{\boldsymbol{\theta} \sim q_t(\boldsymbol{\theta})}\Big[\sum_{i=0}^{n-1} \log p(\mathcal{D}_{t-i} \mid \boldsymbol{\theta})\Big] \Bigg]$$

$$= \underset{q \in \mathcal{Q}}{\arg\min} \ \frac{1-\lambda}{1-\lambda^n} \Bigg[ \sum_{i=0}^{n-1} \lambda^i \mathscr{D}_{KL}(q_t(\boldsymbol{\theta}) \ || \ q_{t-i-1}(\boldsymbol{\theta})) - \mathbb{E}_{\boldsymbol{\theta} \sim q_t(\boldsymbol{\theta})}\Big[ \sum_{i=0}^{n-1} \lambda^i \log p(\mathcal{D}_t \mid \boldsymbol{\theta})$$

$$+ \sum_{i=1}^{n-1} \lambda^i \log p(\mathcal{D}_{t-1} \mid \boldsymbol{\theta}) + \dots + \lambda^{n-1} \log p(\mathcal{D}_{t-n+1} \mid \boldsymbol{\theta}) \Big] \Bigg]$$

$$= \underset{q \in \mathcal{Q}}{\arg\min} \ \frac{1-\lambda}{1-\lambda^n} \Bigg[ \sum_{i=0}^{n-1} \lambda^i \mathscr{D}_{KL}(q_t(\boldsymbol{\theta}) \ || \ q_{t-i-1}(\boldsymbol{\theta})) - \mathbb{E}_{\boldsymbol{\theta} \sim q_t(\boldsymbol{\theta})}\Big[ \frac{1-\lambda^n}{1-\lambda} \log p(\mathcal{D}_t \mid \boldsymbol{\theta})$$

$$+ \frac{\lambda(1-\lambda^{n-1})}{1-\lambda} \log p(\mathcal{D}_{t-1} \mid \boldsymbol{\theta}) + \dots + \lambda^{n-1} \log p(\mathcal{D}_{t-n+1} \mid \boldsymbol{\theta}) \Big] \Bigg]$$

$$= \underset{q \in \mathcal{Q}}{\arg\max} \ \mathbb{E}_{\boldsymbol{\theta} \sim q_t(\boldsymbol{\theta})}\Big[ \sum_{i=0}^{n-1} \frac{\lambda^i(1-\lambda^{n-i})}{1-\lambda^n} \log p(\mathcal{D}_{t-i} \mid \boldsymbol{\theta}) \Big] - \sum_{i=0}^{n-1} \frac{\lambda^i(1-\lambda)}{1-\lambda^n} \mathscr{D}_{KL}(q_t(\boldsymbol{\theta}) \ || \ q_{t-i-1}(\boldsymbol{\theta})).$$

$$\tag{12}$$

$\square$

## C The connection of TD Targets in TD-VCL and Reinforcement Learning

In the Section 4, we formalize the concept of n-Step Temporal-Difference for the Variational CL objective (Definition 4.3). In this Section, we reveal the connections between this definition and the widely used Temporal-Difference methods in Reinforcement Learning. Our aim is to clarify why Equation 6 indeed represents a temporal-difference target, both in a broad and strict senses.

In a **broad** sense, *bootstrapping* characterizes a Temporal-Difference target: building a learning target estimate based on previous estimates. Crucially, the leveraged estimates are functions of different timesteps. TD-VCL objectives applies bootstrapping in the KL regularization term, by considering one or more of posteriors estimates from previous timesteps.

In a **strict** sense, we can show that Equation 6 deeply resembles TD targets in Reinforcement Learning. RL assumes the formalism of a Markov Decision Process (MDP), defined by a tuple $\mathcal{M} = (\mathcal{S}, \mathcal{A}, \mathcal{P}, \mathcal{R}, \mathcal{P}_0, \gamma, H)$, where $\mathcal{S}$ is a state space, $\mathcal{A}$ is an action space, $\mathcal{P} : \mathcal{S} \times \mathcal{A} \times \mathcal{S} \to [0, \infty)$ is a transition dynamics, $\mathcal{R} : \mathcal{S} \times \mathcal{A} \to [-R_{max}, R_{max}]$ is a bounded reward function, $\mathcal{P}_0 : \mathcal{S} \to [0, \infty)$ is an initial state distribution, $\gamma \in [0, 1]$ is a discount factor, and $H$ is the horizon.

The standard RL objective is to find a policy that maximizes the cumulative reward:

$$\pi_{\boldsymbol{\theta}}^* = \arg\max_{\pi} \mathbb{E}_{\pi}[\sum_{k=0}^{H} \gamma^k \mathcal{R}(s_{t+k}, a_{t+k})], \tag{13}$$

with $a_t \sim \pi_{\boldsymbol{\theta}}(a_t \mid s_t)$, $s_t \sim \mathcal{P}(s_t \mid s_{t-1}, a_{t-1})$, and $s_0 \sim \mathcal{P}_0(s)$, where $\pi_{\boldsymbol{\theta}} : \mathcal{S} \times \mathcal{A} \to [0, \infty)$ is a policy parameterized by $\boldsymbol{\theta}$. Hence, we can define the following learning target, which represents a "value" function at each state $s_t$:

$$v_{\pi}(s_t) := \mathbb{E}_{\pi}[\sum_{k=0}^{H} \gamma^k \mathcal{R}(s_{t+k}, a_{t+k}) \mid s = s_t], \forall s_t \in \mathcal{S}. \tag{14}$$

Naturally, it follows that $\pi_{\boldsymbol{\theta}}^* = \arg\max_{\pi} v_{\pi}(s), \forall s \in \mathcal{S}$. Crucially, we can expand Equation 14 as follows:

$$v_{\pi}(s_t) := \mathbb{E}_{\pi}[\sum_{k=0}^{H} \gamma^k \mathcal{R}(s_{t+k}, a_{t+k}) \mid s = s_t]$$

$$= \mathbb{E}_{\pi}[\mathcal{R}(s_t, a_t) + \sum_{k=1}^{H} \gamma^k \mathcal{R}(s_{t+k}, a_{t+k}) \mid s = s_t]$$

$$= \mathbb{E}_{\pi}[\mathcal{R}(s_t, a_t) + \gamma v_{\pi}(s_{t+1})],$$

$$= \mathbb{E}_{\pi}[\mathcal{R}(s_t, a_t) + \gamma \mathcal{R}(s_{t+1}, a_{t+1}) + \gamma^2 v_{\pi}(s_{t+2})],$$

$$= \mathbb{E}_{\pi}[\sum_{k=0}^{n-1} \gamma^k \mathcal{R}(s_t, a_t) + \gamma^n v_{\pi}(s_{t+n})], \forall s_t \in \mathcal{S}, n \leq H. \tag{15}$$

Temporal-Difference methods estimates a learning target directly from Equation 15:

$$\hat{v}_{\pi}(s) := \text{TD}_{\text{RL}}(n) = \underbrace{\mathbb{E}_{\pi}[\sum_{k=0}^{n-1} \gamma^k \mathcal{R}(s_t, a_t)]}_{\text{Estimated via MC Sampling}} + \underbrace{\gamma^n \hat{v}_{\pi}(s_{t+n})}_{\text{Bootstrapped via past estimations}}, \forall s_t \in \mathcal{S}, n \leq H. \tag{16}$$

Now, we turn our attention back to our Variational Continual Learning setting. The standard VCL objective is given by Equation 2:

$$q_t(\boldsymbol{\theta}) = \underset{q \in \mathcal{Q}}{\arg\min} \, \mathscr{D}_{KL}(q(\boldsymbol{\theta}) \, || \, \frac{1}{Z_t} q_{t-1}(\boldsymbol{\theta}) p(\mathcal{D}_t \mid \boldsymbol{\theta})).$$

We can similarly define a learning target as a "value" function which we aim to maximize:

$$
\begin{aligned}
u_{q(\boldsymbol{\theta})}(t) &:= -\mathscr{D}_{KL}(q(\boldsymbol{\theta}) \, || \, \frac{1}{Z_t} q_{t-1}(\boldsymbol{\theta}) p(\mathcal{D}_t \mid \boldsymbol{\theta})) \\
&= \mathbb{E}_{\boldsymbol{\theta} \sim q_t(\boldsymbol{\theta})} \left[ \log p(\mathcal{D}_t \mid \boldsymbol{\theta})] + \log Z_t \right] - \mathscr{D}_{KL}(q_t(\boldsymbol{\theta}) \, || \, q_{t-1}(\boldsymbol{\theta})) \\
&= \mathbb{E}_{\boldsymbol{\theta} \sim q_t(\boldsymbol{\theta})} \left[ \log p(\mathcal{D}_t \mid \boldsymbol{\theta})] + \log Z_t \right] - \mathscr{D}_{KL}(q_t(\boldsymbol{\theta}) \, || \, \frac{1}{Z_{t-1}} q_{t-2}(\boldsymbol{\theta}) p(\mathcal{D}_{t-1} \mid \boldsymbol{\theta})) \\
&= \mathbb{E}_{\boldsymbol{\theta} \sim q_t(\boldsymbol{\theta})} \left[ \log p(\mathcal{D}_t \mid \boldsymbol{\theta})] + \log Z_t \right] + u_{q(\boldsymbol{\theta})}(t-1) \\
&= \mathbb{E}_{\boldsymbol{\theta} \sim q_t(\boldsymbol{\theta})} \left[ \sum_{i=0}^{n-2} \log p(\mathcal{D}_{t-i} \mid \boldsymbol{\theta})] + \sum_{i=0}^{n-2} \log Z_{t-i} \right] + u_{q(\boldsymbol{\theta})}(t-n+1), n \in \mathbb{N}_0, n \le t.
\end{aligned}
\tag{17}
$$

Similarly to the RL case, it follows that $q_t(\boldsymbol{\theta}) = \arg\max_{q \in \mathcal{Q}} u_{q(\boldsymbol{\theta})}(t)$. Lastly, we assume the following estimation of the "value" function defined in Equation 17:

$$
\begin{aligned}
\hat{u}_{q(\boldsymbol{\theta})}(t) &= \mathbb{E}_{\boldsymbol{\theta} \sim q_t(\boldsymbol{\theta})} \left[ \sum_{i=0}^{n-2} \log p(\mathcal{D}_{t-i} \mid \boldsymbol{\theta})] + \sum_{i=0}^{n-2} \log Z_{t-i} \right] + \hat{u}_{q(\boldsymbol{\theta})}(t-n+1) \\
&= \underbrace{\mathbb{E}_{\boldsymbol{\theta} \sim q_t(\boldsymbol{\theta})} \left[ \sum_{i=0}^{n-1} \log p(\mathcal{D}_{t-i} \mid \boldsymbol{\theta})] \right]}_{\text{Estimated via MC Sampling}} - \underbrace{\mathscr{D}_{KL}(q_t(\boldsymbol{\theta}) \, || \, q_{t-n}(\boldsymbol{\theta}))}_{\text{Bootstrapped via past posterior estimations}} + \underbrace{\left[ \sum_{i=0}^{n-1} \log Z_{t-i} \right]}_{\text{Constant w.r.t } \boldsymbol{\theta}}.
\end{aligned}
\tag{18}
$$

We notice that $Z_t$ is constant with respect to $\boldsymbol{\theta}$, hence we can disregard it and still have the same learning target. Thus, we have:

$$
\begin{aligned}
q_t(\boldsymbol{\theta}) &= \underset{q \in \mathcal{Q}}{\arg\max} \, \hat{u}_{q(\boldsymbol{\theta})}(t) \\
&= \underset{q \in \mathcal{Q}}{\arg\max} \, \mathbb{E}_{\boldsymbol{\theta} \sim q_t(\boldsymbol{\theta})} \left[ \sum_{i=0}^{n-1} \log p(\mathcal{D}_{t-i} \mid \boldsymbol{\theta})] \right] - \mathscr{D}_{KL}(q_t(\boldsymbol{\theta}) \, || \, q_{t-n}(\boldsymbol{\theta})) + \left[ \sum_{i=0}^{n-1} \log Z_{t-i} \right] \\
&= \underset{q \in \mathcal{Q}}{\arg\max} \, \underbrace{\mathbb{E}_{\boldsymbol{\theta} \sim q_t(\boldsymbol{\theta})} \left[ \sum_{i=0}^{n-1} \log p(\mathcal{D}_{t-i} \mid \boldsymbol{\theta})] \right] - \mathscr{D}_{KL}(q_t(\boldsymbol{\theta}) \, || \, q_{t-n}(\boldsymbol{\theta}))}_{\text{TD}_{\text{CL}}(n)}.
\end{aligned}
\tag{19}
$$

Equation 19 is exactly n-Step Temporal-Difference target in Definition 4.3 from Section 4. The main differences from the CL recursion in Equation 17 and the RL one in Equation 15 are two-fold. First, the CL setup is not discounted (or, equivalently, assumes the discount factor $\gamma = 1$). Second, the RL recursion looks over future timesteps, while the CL one looks over past timesteps. Besides these two differences, both scenarios are strongly connected. Particularly, they share the same purpose for leveraging TD targets: to strike a balance between MC estimation (which incurs variance) and bootstrapping (which incurs bias) while estimating the learning objective.

# D TD($\lambda$)-VCL is a discounted sum of n-Step TD targets

In Section 4, we mention that the TD-VCL learning target is a compound update that averages n-step temporal-difference targets, as per Proposition 4.4, which we prove below.

**Proposition 4.4.** $\forall n \in \mathbb{N}_0$, $n \leq t$, the objective in Equation 2 can be equivalently represented as:

$$q_t(\boldsymbol{\theta}) = \arg\max_{q \in \mathcal{Q}} \text{TD}_t(n),\tag{7}$$

with $\text{TD}_t(n)$ as in Definition 4.3. Furthermore, the objective in Equation 5 can also be represented as:

$$q_t(\boldsymbol{\theta}) = \arg\max_{q \in \mathcal{Q}} \frac{1-\lambda}{1-\lambda^n} \underbrace{\left[\sum_{k=0}^{n-1} \lambda^k \text{TD}_t(k+1))\right]}_{\text{Discounted sum of TD targets}}.\tag{8}$$

*Proof.* We start by proving the equivalence between Equation 2 and Equation 7:

$$
\begin{aligned}
q_t(\boldsymbol{\theta}) &= \arg\min_{q \in \mathcal{Q}} \mathscr{D}_{KL}(q(\boldsymbol{\theta}) \,\|\, \frac{1}{Z_t} q_{t-1}(\boldsymbol{\theta}) p(\mathcal{D}_t \mid \boldsymbol{\theta})) \\
&= \arg\min_{q \in \mathcal{Q}} \mathscr{D}_{KL}(q(\boldsymbol{\theta}) \,\|\, \frac{1}{\prod_{i=0}^{n-1} Z_{t-i}} q_{t-n}(\boldsymbol{\theta}) \prod_{i=0}^{n-1} p(\mathcal{D}_{t-i} \mid \boldsymbol{\theta})) \\
&= \arg\max_{q \in \mathcal{Q}} \mathbb{E}_{\boldsymbol{\theta} \sim q_t(\boldsymbol{\theta})}\left[\sum_{i=0}^{n-1} \log p(\mathcal{D}_{t-i} \mid \boldsymbol{\theta})\right] - \mathscr{D}_{KL}(q_t(\boldsymbol{\theta}) \,\|\, q_{t-n}(\boldsymbol{\theta})) \\
&= \arg\max_{q \in \mathcal{Q}} \text{TD}_t(n).
\end{aligned}\tag{20}
$$

Now, we show that Equation 5 is a discounted sum of n-Step targets:

$$
\begin{aligned}
q_t(\boldsymbol{\theta}) &= \arg\max_{q \in \mathcal{Q}} \frac{1-\lambda}{1-\lambda^n} \Bigg[ \mathbb{E}_{\boldsymbol{\theta} \sim q_t(\boldsymbol{\theta})}[\log p(\mathcal{D}_t \mid \boldsymbol{\theta}) - \mathscr{D}_{KL}(q_t(\boldsymbol{\theta}) \,\|\, q_{t-1}(\boldsymbol{\theta}))] \\
&\qquad + \lambda \mathbb{E}_{\boldsymbol{\theta} \sim q_t(\boldsymbol{\theta})}[\log p(\mathcal{D}_t \mid \boldsymbol{\theta}) + \log p(\mathcal{D}_{t-1} \mid \boldsymbol{\theta})] - \lambda \mathscr{D}_{KL}(q_t(\boldsymbol{\theta}) \,\|\, q_{t-2}(\boldsymbol{\theta})) + \ldots \\
&\qquad + \lambda^{n-1} \mathbb{E}_{\boldsymbol{\theta} \sim q_t(\boldsymbol{\theta})}[\sum_{i=0}^{n-1} \log p(\mathcal{D}_{t-i} \mid \boldsymbol{\theta})] - \lambda^{n-1} \mathscr{D}_{KL}(q_t(\boldsymbol{\theta}) \,\|\, q_{t-n}(\boldsymbol{\theta})) \Bigg] \\
&= \arg\max_{q \in \mathcal{Q}} \frac{1-\lambda}{1-\lambda^n} \left[\text{TD}_t(1) + \lambda \text{TD}_t(2) + \ldots \lambda^{n-1} \text{TD}_t(n)\right] \\
&= \arg\max_{q \in \mathcal{Q}} \frac{1-\lambda}{1-\lambda^n} \underbrace{\left[\sum_{k=0}^{n-1} \lambda^k \text{TD}_t(k+1))\right]}_{\text{Disconted sum of TD targets}}.
\end{aligned}
$$

$$\tag{21}$$

$\square$

In Equation 7, if we set $n = 1$, the n-Step TD target recovers the VCL objective. Furthermore, it is worth highlighting that an n-Step TD target is **not** the same as n-Step KL Regularization. The latter leverages several previous posterior estimates, while the former only relies on a single estimate. Lastly, we can follow a similar idea to prove that the n-Step KL Regularization objective is a simple average of n-step TD targets, by leveraging the expansion in Equation 9 and identifying the sum of TD targets.

# E   TD-VCL: A spectrum of Continual Learning algorithms

In this Section, we describe how TD-VCL spans a spectrum of algorithms that mix different levels of Monte Carlo approximation for expected log-likelihood and KL regularization. Our goal is to show that by choosing specific hyperparameters for Equation 5, one may recover vanilla VCL in one extreme and n-Step KL regularization in the opposite.

Let us consider the TD-VCL objective in Equation 5:

$$\arg\max_{q\in\mathcal{Q}} \mathbb{E}_{\boldsymbol{\theta}\sim q_t(\boldsymbol{\theta})}\Big[ \sum_{i=0}^{n-1} \frac{\lambda^i(1-\lambda^{n-i})}{1-\lambda^n} \log p(\mathcal{D}_{t-i} \mid \boldsymbol{\theta})\Big] - \sum_{i=0}^{n-1} \frac{\lambda^i(1-\lambda)}{1-\lambda^n} \mathscr{D}_{KL}(q_t(\boldsymbol{\theta}) \mid\mid q_{t-i-1}(\boldsymbol{\theta})).$$

Trivially, if we set $\lambda = 0$, assuming $0^0 = 1$, it recovers the Vanilla VCL objective, as stated in Equation 3, regardless of the choice of $n$.

More interestingly, we investigate the learning target as $\lambda \to 1$:

$$\lim_{\lambda\to 1} \left\{ \mathbb{E}_{\boldsymbol{\theta}\sim q_t(\boldsymbol{\theta})}\Big[ \sum_{i=0}^{n-1} \frac{\lambda^i(1-\lambda^{n-i})}{1-\lambda^n} \log p(\mathcal{D}_{t-i} \mid \boldsymbol{\theta})\Big] - \sum_{i=0}^{n-1} \frac{\lambda^i(1-\lambda)}{1-\lambda^n} \mathscr{D}_{KL}(q_t(\boldsymbol{\theta}) \mid\mid q_{t-i-1}(\boldsymbol{\theta})) \right\}$$

$$= \mathbb{E}_{\boldsymbol{\theta}\sim q_t(\boldsymbol{\theta})}\Big[ \sum_{i=0}^{n-1} \underbrace{\lim_{\lambda\to 1}\left\{\frac{\lambda^i(1-\lambda^{n-i})}{1-\lambda^n}\right\}}_{(\mathrm{I})} \log p(\mathcal{D}_{t-i} \mid \boldsymbol{\theta})\Big] - \sum_{i=0}^{n-1} \underbrace{\lim_{\lambda\to 1}\left\{\frac{\lambda^i(1-\lambda)}{1-\lambda^n}\right\}}_{(\mathrm{II})} \mathscr{D}_{KL}(q_t(\boldsymbol{\theta}) \mid\mid q_{t-i-1}(\boldsymbol{\theta}))$$

Let us develop (I) and (II) separately by applying the L'Hôpital's rule. First, for (I):

$$\begin{aligned}
\lim_{\lambda\to 1}\left\{\frac{\lambda^i(1-\lambda^{n-i})}{1-\lambda^n}\right\} &= \lim_{\lambda\to 1}\left\{\frac{i\lambda^{i-1}(1-\lambda^{n-i}) - \lambda^i(n-i)\lambda^{n-i-1}}{-n\lambda^{n-1}}\right\} \\
&= \lim_{\lambda\to 1}\left\{\frac{i\lambda^{i-1} - i\lambda^{n-1} - (n-i)\lambda^{n-1}}{-n\lambda^{n-1}}\right\} = \frac{n-i}{n}.
\end{aligned} \tag{22}$$

Now, for (II):

$$\lim_{\lambda\to 1}\left\{\frac{\lambda^i(1-\lambda)}{1-\lambda^n}\right\} = \lim_{\lambda\to 1}\left\{\frac{i\lambda^{i-1}(1-\lambda) - \lambda^i}{-n\lambda^{n-1}}\right\} = \frac{1}{n}. \tag{23}$$

Applying Equations 22 and 23 to TD-VCL objective, we obtain:

$$\arg\max_{q\in\mathcal{Q}} \mathbb{E}_{\boldsymbol{\theta}\sim q_t(\boldsymbol{\theta})}\Big[ \sum_{i=0}^{n-1} \frac{(n-i)}{n} \log p(\mathcal{D}_{t-i} \mid \boldsymbol{\theta})\Big] - \sum_{i=0}^{n-1} \frac{1}{n} \mathscr{D}_{KL}(q_t(\boldsymbol{\theta}) \mid\mid q_{t-i-1}(\boldsymbol{\theta})),$$

which is exactly the N-Step KL Regularization objective.

## F  Impact Statement

This work develops a novel learning objective for Bayesian Continual Learning. As such, we believe our work has a positive impact on fundamental research for Machine Learning for three reasons. First, we argue that advancing Continual Learning research is crucial for ensuring the long-term quality of ML models in production systems, as they are vulnerable to potential distributional shifts in the data generation distribution. We also argue that CL is crucial for developing safe autonomous learning agents, as Catastrophic Forgetting may be a dangerous challenge while interacting with the physical or digital world. Second, our particular focus on the Bayesian framework is relevant for designing uncertainty-aware models, which, as argued in Section 1, is crucial for robust Machine Learning and general AI safety. Lastly, we provide a solid theoretical connection between Variational Continual Learning methods and Temporal-Difference methods, effectively bridging two seemingly distant disciplines into a unified family of algorithms. We believe this will inspire further research in the intersection of both areas.

## G  Implementation Details and Reproducibility

**Operationalization.** For all experiments, we use a Gaussian mean-field approximate posterior and assume a Gaussian prior $\mathcal{N}(0, \sigma^2 \boldsymbol{I})$ for the variational methods. We parameterize all distributions as deep networks. For all considered objectives, we compute the KL term analytically and employ the Monte Carlo approximations for the expected log-likelihood terms, leveraging the reparametrization trick [47] for computing gradients. Lastly, we employ likelihood-tempering [33] to prevent variational over-pruning [48].

**Model Architecture and Hyperpatameters**. We adopt fully connected neural networks for PermutedMNIST-Hard, SplitMNIST-Hard and SplitNotMNIST-Hard. We choose different depths and sizes depending on the benchmark, and we provide a full list of hyperparameters in Appendix H. For CIFAR100-10 and TinyImageNet-10, we implement a Bayesian version of the AlexNet [56], a traditional convolutional neural network architecture, as in prior Bayesian CL literature [39]. Crucially, also following prior literature [38], we do not use pre-trained representations, as our goal is to evaluate how the proposed objectives perform in the CL setting, which also requires learning their own robust representations. Finally, for training, we adopt the Adam optimizer [57] and employ early stopping with a patience parameter of five epochs, which drastically reduces the number of epochs needed for each new task in comparison to previous work [13].

**Hyperparamter Tuning Protocol.** We conduct hyperparameter tuning for all methods in the paper, including the baselines (VCL, UCL, UCB). We follow a random search for each evaluated benchmark. For a fair comparison, we ensure that all methods use approximately the same compute of 1 GPU day. We provide the search space for each method in our released code. For the proposed methods, we mainly tuned three hyperparameters: $n$ (as in n-Step KL), $\lambda$ (as in TD-VCL), and $\beta$ (the likelihood tempering parameter). We conducted a grid search for each evaluated benchmark, with $n \in \{1, 2, 3, 5, 8, 10\}$, $\lambda \in \{0.0, 0.1, 0.5, 0.8, 0.9, 0.99\}$, and $\beta \in \{1e-5, 1e-4, 1e-3, 5e-3, 1e-2, 5e-2, 1e-1, 1.0\}$.

**Reproducibility**. Reported results are averaged across ten different seeds for PermutedMNIST-Hard, SplitMNIST-Hard, and SplitNotMNIST-Hard, and five seeds for CIFAR100-10 and TinyImageNet-10. Error bars represent 95% confidence intervals, while tables show 2-sigma errors up to two decimal places. We execute all experiments using a single GPU RTX 4090. We provide our implementation code for the proposed methods (TD-VCL, TD-UCB, TD-UCL, and n-Step), as well as considered baselines (Batch MLE, Online MLE, VCL, VCL CoreSet, UCB, and UCL) in `https://github.com/luckeciano/TD-VCL`.

# H  Hyperparameters

Table 4 provides the shared hyperparameters used in each benchmark. Tables 5 and 6 provided the specific hyperparameters for the proposed methods and baselines, respectively.

| | PermMNIST-Hard | SplitMNIST-Hard | SplitNotMNIST-Hard | CIFAR100-10 | TinyImageNet-10 |
|---|---|---|---|---|---|
| **Batch Size** | 256 | 256 | 256 | 256 | 256 |
| **Max Epochs** | 100 | 100 | 100 | 100 | 100 |
| **NN Architecture** | [100, 100] | [256, 256] | [150, 150, 150, 150] | AlexNet | AlexNet |
| **Number of Heads** | 1 | 1 | 1 | 10 | 10 |
| **Learning Rate** | 1e-3 | 1e-3 | 1e-3 | 1e-3 | 1e-3 |

Table 4: Training hyperparameters. These are shared across all evaluated methods.

| | | PermMNIST-Hard | SplitMNIST-Hard | SplitNotMNIST-Hard | CIFAR100-10 | TinyImageNet-10 |
|---|---|---|---|---|---|---|
| **n-Step KL** | $n$ | 5 | 4 | 5 | 5 | 2 |
| | $\beta$ | 5e-3 | 5e-2 | 5e-2 | 3e-5 | 1e-9 |
| **TD($\lambda$)-VCL** | $n$ | 8 | 4 | 3 | 10 | 2 |
| | $\lambda$ | 0.5 | 0.8 | 0.1 | 0.5 | 0.1 |
| | $\beta$ | 1e-3 | 5e-2 | 1e-3 | 1e-5 | 1e-9 |
| **TD($\lambda$)-UCL** | $n$ | 8 | 4 | 3 | 5 | 2 |
| | $\lambda$ | 0.5 | 0.8 | 0.1 | 0.8 | 0.5 |
| | $\beta$ | 1e-3 | 5e-2 | 1e-3 | 1e-5 | 1e-7 |
| **TD($\lambda$)-UCB** | $n$ | 8 | 4 | 3 | 8 | 3 |
| | $\lambda$ | 0.5 | 0.8 | 0.1 | 0.8 | 0.1 |
| | $\beta$ | 1e-3 | 5e-2 | 1e-3 | 1e-5 | 1e-5 |

Table 5: Hyperparameters for different methods across benchmarks.

| | | PermMNIST-Hard | SplitMNIST-Hard | SplitNotMNIST-Hard | CIFAR100-10 | TinyImageNet-10 |
|---|---|---|---|---|---|---|
| **VCL** | $\beta$ | 5e-3 | 5e-3 | 5e-3 | 5e-4 | 1e-5 |
| **UCL** | $\alpha$ | 1.0 | 10.0 | 0.5 | 1.0 | 10.0 |
| | $\beta$ | 0.001 | 1.0 | 0.001 | 0.001 | 1.0 |
| | $\gamma$ | 0.01 | 1.0 | 1.0 | 0.005 | 0.1 |
| | r | 0.5 | 0.5 | 0.5 | 0.5 | 0.5 |
| | $\beta_{kl}$ | 5e-3 | 1e-3 | 1e-5 | 1e-4 | 1e-7 |
| **UCB** | $\alpha$ | 1.0 | 1.0 | 0.1 | 10.0 | 100.0 |
| | $\beta$ | 1e-2 | 1e-2 | 5e-2 | 5e-5 | 1e-5 |

Table 6: Hyperparameters for different methods across benchmarks.

# I  PermutedMNIST-Hard, SplitMNIST-Hard, and SplitNotMNIST-Hard: Introducing Higher Standards for MNIST/NotMNIST-based Continual Learning Benchmarks

Popular Continual Learning benchmarks, such as PermutedMNIST, SplitMNIST, and SplitNotMNIST, [1, 20, 13] provide an effective experimental setup. These benchmarks offer tasks that, while conceptually simple in isolation, present a challenging task-streaming setup that highlights the phenomenon of Catastrophic Forgetting. This combination facilitates the study of Continual Learning methods through rapid iterations and modest deep architectures, making it ideal for academic settings. Nonetheless, we argue that the "unrestricted" versions of these benchmarks are either trivially addressed by simple baselines or do not reflect a challenging evaluation setup for Catastrophic Forgetting in current Bayesian CL research. This observation motivates our work to incorporate certain restrictions in the considered methods, resulting in a more challenging setup for Continual Learning while maintaining the benchmarks' original desiderata.

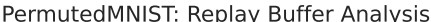

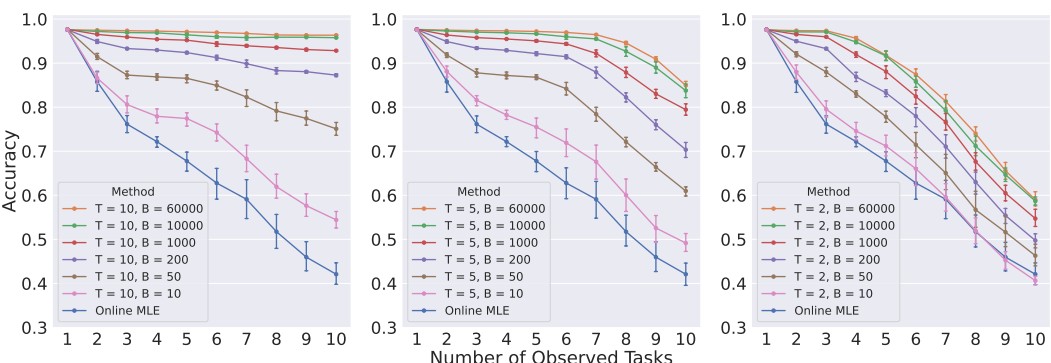

Figure 4: **A Replay Buffer analysis on the PermutedMNIST**. Each curve represents a model re-trained on a buffer composed of "$T$" previous tasks, "$B$" examples of each. Online MLE only considers the current task. Allowing "unlimited" access to previous task data trivializes the CL setting, and a simple MLE baseline is enough to attain strong results. Nevertheless, as we restrict the replay buffer in size and number of tasks, the benchmark becomes substantially more challenging and shows signs of Catastrophic Forgetting.

**Restricting replay memory size imposes a new challenge for MNIST/NotMNIST CL benchmarks**. Figure 4 presents MLE models trained on different levels of previous tasks' data (besides the data from the current task) for the classic PermutedMNIST benchmark. Online MLE means no usage of data from previous tasks. On the flip side, we re-train the remaining models considering the data of $T$ previous tasks, with $B$ examples of each. It shows that allowing access to all the old tasks is enough for an MLE model to maintain high accuracy even when presenting to only a set as tiny as 200 examples. As we reduce the number of old tasks in the buffer, performance decreases, showing clear signs of Catastrophic Forgetting. For $T = 2$, all models present an accuracy lower than 60% regardless of the volume of old task data. Therefore, in order to impose a harder evaluation setup, we impose additional restrictions for re-training in prior tasks. For PermutedMNIST-Hard, we restrict re-training to the two most recent past tasks, with 200 examples per task; for SplitMNIST-Hard and SplitNotMNIST-Hard, we allow only the most recent past task with 40 examples. As shown in Figure 4, MLE-based methods do not perform well in this setting. Crucially, these adopted replay buffers are very small in comparison with the training data of the current task, which is more realistic than retaining the full data. Nonetheless, they strictly follow the core set sizes used in prior work [13], ensuring that the adopted baselines (e.g., VCL CoreSet) work as proposed and promoting a fair comparison.

**"Single-Head" Classifiers prevents the saturation of PermutedMNIST, SplitMNIST, and SplitNotMNIST**. "Multi-Head" networks train a different classifier for each task on top of a shared backbone. The goal is to alleviate Catastrophic Forgetting by disregarding the effect of negative transfer among tasks. While this may be acceptable for harder datasets where multi-head architec-

ture is necessary to avoid trivial performance, current methods with multi-head classifiers already saturates the classic MNIST/NotMNIST benchmarks, achieving accuracy above 99%. For empirical evidence, we evaluate the methods on SplitMNIST (which allows multi-head architecture, Figure 5) and SplitMNIST-Hard (which restricts to a single-head classifier, Figure 6 in Appendix K). In the former, all baselines trivially attain high average accuracy; in the latter, all methods face a much more challenging setup. Hence, PermutedMNIST-Hard, SplitMNIST-Hard, and SplitNotMNIST-Hard enforces single-head architecture.

Figure 5: **SplitMNIST results**. The first five plots show results per task, and the last one is an average across tasks. As a consequence of multi-head networks simplifying the Continual Learning challenge, all methods attain high accuracy. In particular, variational methods accuracies ranging from 97% and 98%. In constrast, SplitMNIST-Hard in Figure 6, provides a considerably more challenging CL benchmark.

Lastly, we highlight that all evaluated methods – including the proposed ones – are subject to the adopted restrictions highlighted in this Section. Therefore, they are trained in the same data with the same parametrization, ensuring a fair comparison setup.

## J   Benchmarks Description

**PermutedMNIST-Hard**. This benchmark uses the MNIST dataset. Each task corresponds to a different permutation of the pixels in the MNIST data. Similarly to MNIST, PermutedMNIST is a multi-class classification problem to recognize the handwritten digit associated with the image. The benchmark runs 10 successive tasks, and each evaluation iteration considers the performance in all past tasks. For the "Hard" version, we restrict any method in two ways, as described in Appendix I: first, replay buffers are restricted to the *two most recent tasks*, with a fixed set of *200 data points per task*; second, we restrict the model architectures to single-head classifiers.

**SplitMNIST-Hard**. This benchmark also considers the MNIST dataset but in a binary classification setting. The model selects between two different digits. Five tasks from the MNIST dataset arrive in sequence: 0/1, 2/3, 4/5, 6/7, and 8/9, and evaluation considers the performance in all past tasks. For the "Hard" version, we apply the similar restrictions: replay buffers restricted to the *most recent task*, with a fixed set of *40 data points*. We also restrict the model architectures to single-head classifiers.

**SplitNotMNIST-Hard**. This benchmark contains a similar structure to SplitMNIST-Hard, but it leverages the notMNIST dataset. This more challenging task contains characters from diverse font styles, comprising 400,000 examples. The five tasks are A/F, B/G, C/H, D/I, and E/J. The "Hard" version applies the same restrictions as in SplitMNIST-Hard.

**CIFAR100-10**. This challenging benchmark contains 10 different tasks, each of them comprising 20 distinct classes from the CIFAR-100 dataset [51]. Evaluation considers the performance in all previous tasks. The dataset contains 50,000 images (5,000 per task) for training/validation and 10,000 images (1,000 per task) for evaluation. For this benchmark, we restrict the replay buffer to contain *200 data points per task*.

**TinyImageNet-10**. This challenging benchmark also contains 10 different tasks, each of them comprising 20 distinct classes from the ImageNet dataset [58]. The dataset contains 100,000 images (10,000 per task) for training/validation and 10,000 images (1,000 per task) for evaluation. Particularly for TinyImageNet-10, we also adopt a memory restriction: replay buffers are restricted to the *three most recent tasks*, with a fixed set of *200 data points per task*.

# K  Per Task Performance: Additional Results

## K.1  SplitMNIST-Hard

Figure 6 presents the per-task performance for the SplitMNIST-Hard results. As expected, the performance of all methods drops substantially in comparison to traditional SplitMNIST, as the CL becomes considerably harder. However, we highlight that n-Step KL and TD-VCL presented better results than VCL and VCL CoreSet, demonstrating again the effectiveness of the proposed learning objectives.

Interestingly, the average accuracy does not decrease monotonically, as one might typically expect due to Catastrophic Forgetting. Instead, it drops significantly after Task 3 and then rises again. This evidence indicates two potential dynamics of transfer learning: a negative transfer from Task 1 while learning Task 3, and a positive transfer from Task 1 while learning Task 4. For instance, the digit "0" from Task 1 is rounded, similar to the digits "5" and "6" in Tasks 3 and 4, respectively. Additionally, the digit "1" is composed of straight lines, much like the digits "4" and "7." We believe that the employed architecture, given its inherent and intended simplicity, relies on features of this nature. Therefore, more expressive architectures that better disentangle these features may potentially prevent the negative transfer. However, exploring this possibility is beyond our scope, as our focus is on studying the effects of Catastrophic Forgetting in Continual Learning.

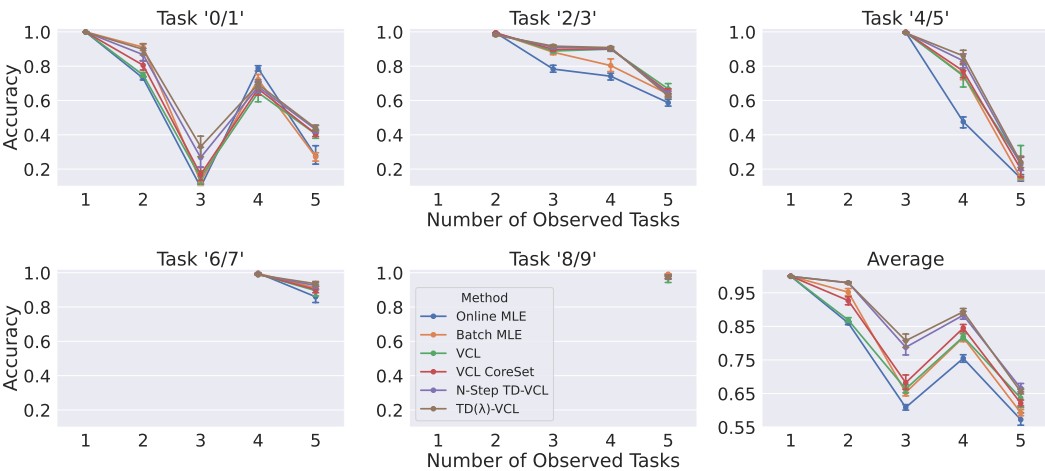

Figure 6: **SplitMNIST-Hard results**. In this more robust evaluation setting, tasks are enforced to share a single classifier with restricted replay memory. Consequently, the effect of Catastrophic Forgetting (and task negative transfer) is explicit. TD-VCL objectives present slightly better average accuracy across tasks in comparison with standard VCL variants.

## K.2  SplitNotMNIST-Hard

In this section, we show per-task performance for SplitNotMNIST-Hard. As highlighted in Section 5.1, NotMNIST is a considerably harder dataset than MNIST, and the choice of simpler deep architectures naturally results in higher approximation errors. Our goal is to evaluate how the presented methods behave under this circumstance.

Figure 7 presents the results. As expected, even learning the current task is challenging. This characteristic contrasts with MNIST-based benchmarks, where all models could at least fit the current task almost perfectly. MLE methods fit the current task slightly better since their objectives are not regularized by the prior or previous posterior. However, this same reason caused them to suffer from Catastrophic Forgetting more drastically, as they tend to focus on fitting the current task and disregard past ones. Overall, TD-VCL objectives maintained the best trade-off between plasticity and memory stability, aligning with the results in the other benchmarks.

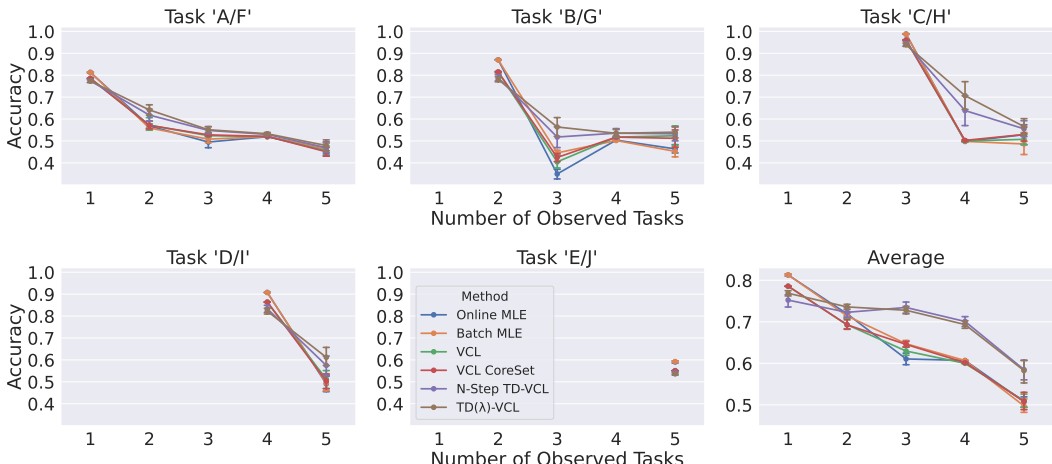

Figure 7: **SplitNotMNIST-Hard results**. The first five plots show results per task, and the last one is an average across them. SplitNotMNIST-Hard is considerably harder to fit with modest deep architectures, leading to a setup where posteriors induce high approximation errors. As a result, the standard VCL variants performs similarly to non-variational approaches. TD-VCL surpasses all methods and shows more robustness to Catastrophic Forgetting under this high approximation error setting.

### K.3 CIFAR100-10

Figure 8 displays the per-task performance in the CIFAR100-10 benchmark. Non-variational baselines consistently struggle with Catastrophic Forgetting, even in more recent tasks. VCL and VCL CoreSet also show a consistent drop in accuracy as the number of observed tasks increases, although this decline is less noticeable in some cases and occasionally followed by a slight increase in accuracy for certain tasks. In contrast, the proposed TD-VCL objectives demonstrate a significant improvement over the baselines and show little indication of Catastrophic Forgetting, despite the harder challenge posed by the CIFAR100 dataset.

Interestingly, variational methods, which experience less Catastrophic Forgetting, exhibit a surprising effect in some tasks: their accuracy initially drops after observing a few consecutive tasks before subsequently increasing again. For example, in Task 3, this effect is evident across all variational methods. As a result, the average accuracy tends to rise as the total number of observed tasks increases, which is also reported in prior work (see Figure 7a in Ahn et al. [43], and Table 2 in Thapa and Li [39])). We hypothesize that the process of explicit posterior regularization, combined with training on successive tasks, leads to a parameterization that learns features more generalizable across tasks, incurring positive transfer learning.

### K.4 TinyImageNet-10

Lastly, Figure 9 illustrates the per-task performance in the TinyImageNet-10 benchmark. As seen in previous scenarios, Online MLE consistently fails to achieve continual learning. Interestingly, VCL also encounters difficulties in this more challenging benchmark, showing per-task performance similar to Batch MLE. VCL CoreSet outperforms the standard VCL and achieves performance comparable to the TD-VCL objectives in some tasks. Nevertheless, the TD-VCL objectives consistently demonstrate superior performance across all tasks, reinforcing the findings from the earlier benchmarks.

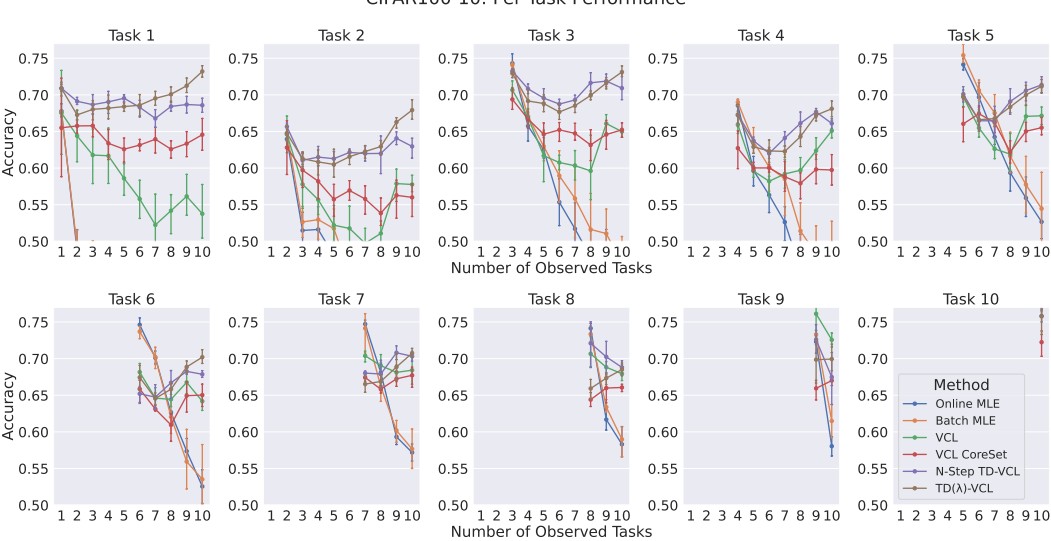

Figure 8: **Per-task performance (accuracy) over time in the CIFAR100-10 benchmark**. Each plot illustrates the accuracy of a specific task (as indicated in the plot title) as the number of observed tasks increases. Non-variational baselines consistently struggle with catastrophic forgetting, while VCL and VCL CoreSet show a mild effect. However, the TD-VCL objectives demonstrate a noticeable improvement over these methods, even in the more challenging setup.

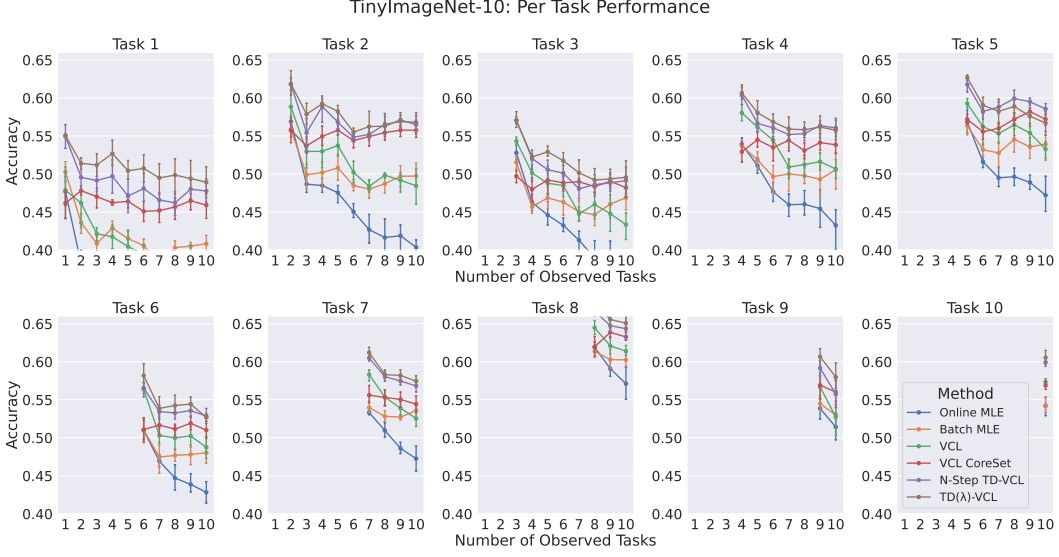

Figure 9: **Per-task performance over time in the TinyImageNet-10 benchmark.**. In the most challenging benchmark presented in this work, we observe similar trends to the previous ones, where TD-VCL objectives show superior performance across tasks.

# L   Hyperparameters Robustness Analysis

In this Section, we present robustness studies in the PermutedMNIST-Hard benchmark with respect to the relevant hyperparameters. Our goal is to evaluate how they affect the performance of the proposed methods.

## L.1   n-Step KL Regularization

Figure 10 presents the ablation study of the n-step KL Regularization method in the PermutedMNIST-Hard benchmark. We designed this study to highlight the two most sensitive hyperparameters: $n$, the n-step size, and $\beta$, the likelihood-tempering parameter.

Similarly to VCL, this method is sensitive to the choice of $\beta$. Higher values will prevent the model from fitting new tasks, a manifestation of variational over-pruning. On the other hand, lower values will not retain knowledge properly, suffering from Catastrophic Forgetting. Mild values (0.001, 0.005, 0.01) balanced well this trade-off.

In terms of $n$, we observe benefits of up to 5 steps. Beyond that, the effect saturates, even becoming slightly detrimental. This observation suggests the existence of an optimal range for $n$ while leveraging past posterior estimates.

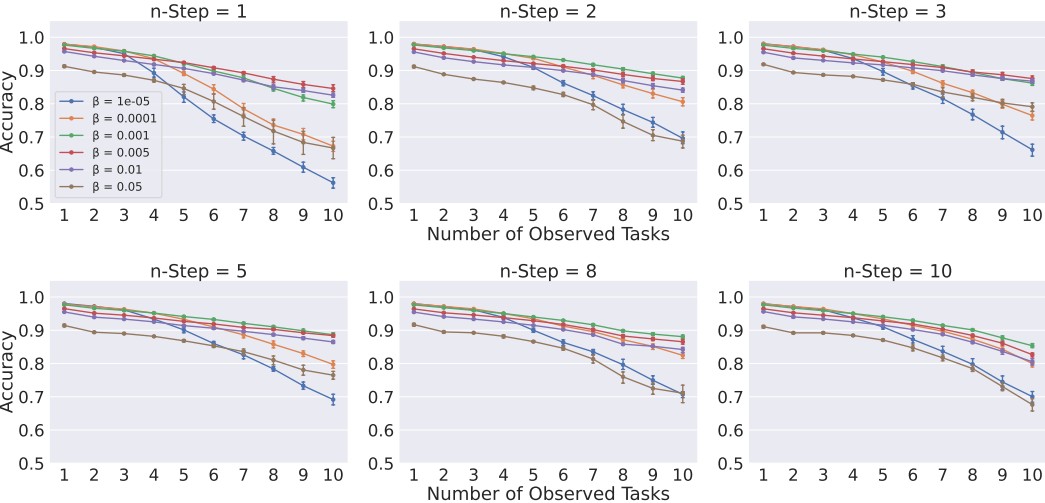

Figure 10: **Hyperparameter Robustness Analysis for n-Step KL Regularization in PermutedMNIST-Hard.** The plots show the effect of the likelihood-tempering parameter $\beta$ for different $n$. For $\beta$, too high values negatively affect fitting new tasks, and too low values disregard the regularization of previous posteriors, leading to Catastrophic Forgetting. For $n$, we observe benefits while increasing up to $n = 5$, and the effect saturates.

## L.2   TD($\lambda$)-VCL

Figure 11 shows the ablation study for TD-VCL. For this setup, we considered a fixed value of $\beta$, as our hyperparameter search suggested the same trends for n-Step KL Regularization and TD-VCL. Hence, we simplify the analysis to consider only $n$ and $\lambda$.

TD-VCL presents mild sensitivity to the choice of $\lambda$. The effect is more pronounced as the method observes more tasks, with a slight preference for lower values for some choices of $n$. We believe that the choice of $\lambda$ will fundamentally depend on how most recent estimates are better and more informative than old ones. In the case where they present similar approximation errors, the choice of $\lambda$ causes less impact, and, therefore, there is less difference between leveraging N-Step TD-VCL and TD($\lambda$)-VCL objectives.

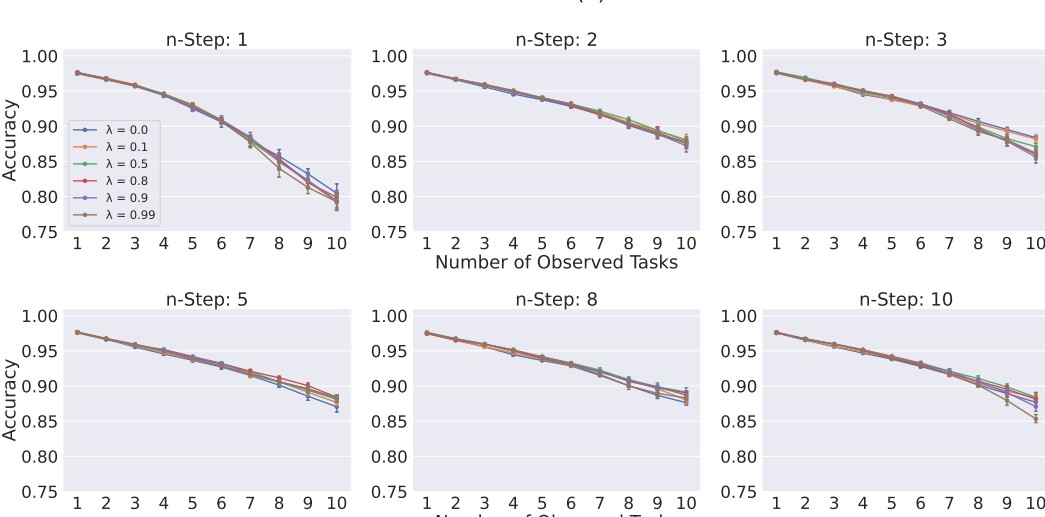

Figure 11: **Hyperparameter Robustness Analysis for TD($\lambda$)-VCL in PermutedMNIST-Hard**. The plots show the effect of $\lambda$ for different choices of $n$. The learning objective presents mild sensitivity to the choice of $\lambda$ in this benchmark, and the effect is more pronounced as the number of observed tasks increases.

## M   Full Table Results

In this Appendix, we report the full version of Tables 1 and 3, for the sake of completeness. Table 7 shows the results on CIFAR100-10 and TinyImageNet-10, considering all timesteps from $t = 2$ to $t = 10$. Table 8 shows the results for all benchmarks, including SplitNotMNIST-Hard, for the Bayesian CL methods and their TD-enhanced counterparts.

Table 7: **Full table for quantitative comparison on the CIFAR100-10 and TinyImagenet-10 benchmarks**. Each column presents the average accuracy across the past t observed tasks. Results are reported with two standard deviations across five seeds. TD-VCL variants consistently outperform the baselines in harder benchmarks with more complex architectures, such as Bayesian CNNs.

| | **CIFAR100-10** | | | | | | | | |
| | t = 2 | t = 3 | t = 4 | t = 5 | t = 6 | t = 7 | t = 8 | t = 9 | t = 10 |
|---|---|---|---|---|---|---|---|---|---|
| Online MLE | 0.56±0.05 | 0.56±0.06 | 0.57±0.06 | 0.56±0.04 | 0.56±0.03 | 0.55±0.03 | 0.53±0.06 | 0.51±0.04 | 0.52±0.04 |
| Batch MLE | 0.57±0.03 | 0.58±0.04 | 0.58±0.04 | 0.59±0.04 | 0.58±0.05 | 0.58±0.06 | 0.56±0.06 | 0.54±0.05 | 0.54±0.07 |
| VCL | 0.64±0.02 | 0.63±0.03 | 0.63±0.02 | 0.60±0.02 | 0.60±0.02 | 0.60±0.03 | 0.61±0.05 | 0.65±0.02 | 0.66±0.01 |
| VCL CoreSet | 0.64±0.05 | 0.65±0.03 | 0.63±0.03 | 0.62±0.03 | 0.63±0.02 | 0.63±0.02 | 0.61±0.02 | 0.64±0.03 | 0.65±0.02 |
| **n-Step TD-VCL** | 0.67±0.01 | 0.68±0.01 | **0.67±0.02** | **0.67±0.01** | **0.65±0.01** | **0.66±0.01** | **0.68±0.04** | **0.69±0.01** | **0.69±0.02** |
| **TD($\lambda$)-VCL** | **0.66±0.02** | **0.67±0.02** | **0.66±0.04** | **0.66±0.01** | **0.66±0.02** | **0.66±0.01** | **0.67±0.01** | **0.69±0.02** | **0.71±0.01** |

| | **TinyImagenet-10** | | | | | | | | |
| | t = 2 | t = 3 | t = 4 | t = 5 | t = 6 | t = 7 | t = 8 | t = 9 | t = 10 |
|---|---|---|---|---|---|---|---|---|---|
| Online MLE | 0.48±0.03 | 0.45±0.02 | 0.45±0.02 | 0.46±0.02 | 0.44±0.01 | 0.44±0.02 | 0.45±0.02 | 0.45±0.02 | 0.44±0.03 |
| Batch MLE | 0.50±0.02 | 0.47±0.02 | 0.48±0.02 | 0.49±0.02 | 0.48±0.02 | 0.48±0.02 | 0.50±0.02 | 0.50±0.02 | 0.51±0.03 |
| VCL | 0.53±0.06 | 0.50±0.02 | 0.51±0.03 | 0.52±0.02 | 0.51±0.03 | 0.49±0.01 | 0.51±0.02 | 0.51±0.02 | 0.51±0.02 |
| VCL CoreSet | 0.52±0.03 | 0.50±0.02 | 0.51±0.02 | 0.53±0.01 | 0.51±0.02 | 0.52±0.01 | 0.54±0.02 | 0.55±0.02 | 0.54±0.02 |
| **n-Step TD-VCL** | **0.56±0.02** | 0.54±0.03 | 0.55±0.02 | 0.55±0.02 | 0.54±0.02 | 0.54±0.01 | 0.56±0.02 | 0.56±0.01 | 0.56±0.02 |
| **TD($\lambda$)-VCL** | **0.57±0.03** | **0.55±0.02** | **0.56±0.02** | **0.56±0.01** | **0.55±0.03** | **0.55±0.03** | **0.56±0.02** | **0.57±0.02** | **0.56±0.02** |

Table 8: **Full table for quantitative comparison between Bayesian CL methods and their TD-enhanced counterparts**. The TD-enhanced methods incorporate the objective in Equation 5 in each base method. Although no single base method consistently outperforms the others across all benchmarks, their TD-enhanced versions consistently achieve better performance, particularly at later timesteps.

**PermutedMNIST-Hard**

| | t = 2 | t = 3 | t = 4 | t = 5 | t = 6 | t = 7 | t = 8 | t = 9 | t = 10 |
|---|---|---|---|---|---|---|---|---|---|
| VCL | $0.95_{\pm0.00}$ | $0.94_{\pm0.01}$ | $0.93_{\pm0.02}$ | $0.91_{\pm0.02}$ | $0.89_{\pm0.03}$ | $0.86_{\pm0.03}$ | $0.83_{\pm0.04}$ | $0.80_{\pm0.06}$ | $0.78_{\pm0.04}$ |
| **TD($\lambda$)-VCL** | $\mathbf{0.97_{\pm0.00}}$ | $\mathbf{0.96_{\pm0.00}}$ | $\mathbf{0.95_{\pm0.00}}$ | $\mathbf{0.94_{\pm0.01}}$ | $\mathbf{0.93_{\pm0.01}}$ | $\mathbf{0.92_{\pm0.01}}$ | $\mathbf{0.91_{\pm0.01}}$ | $\mathbf{0.90_{\pm0.01}}$ | $\mathbf{0.89_{\pm0.02}}$ |
| UCL | $0.97_{\pm0.00}$ | $0.95_{\pm0.01}$ | $0.94_{\pm0.01}$ | $0.92_{\pm0.02}$ | $0.89_{\pm0.02}$ | $0.86_{\pm0.04}$ | $0.83_{\pm0.06}$ | $0.78_{\pm0.09}$ | $0.73_{\pm0.12}$ |
| **TD($\lambda$)-UCL** | $\mathbf{0.97_{\pm0.00}}$ | $\mathbf{0.97_{\pm0.00}}$ | $\mathbf{0.95_{\pm0.00}}$ | $\mathbf{0.94_{\pm0.01}}$ | $\mathbf{0.92_{\pm0.02}}$ | $\mathbf{0.90_{\pm0.02}}$ | $\mathbf{0.88_{\pm0.04}}$ | $\mathbf{0.85_{\pm0.09}}$ | $\mathbf{0.84_{\pm0.04}}$ |
| UCB | $0.93_{\pm0.01}$ | $0.93_{\pm0.01}$ | $0.92_{\pm0.01}$ | $0.90_{\pm0.01}$ | $0.89_{\pm0.02}$ | $0.87_{\pm0.02}$ | $0.86_{\pm0.02}$ | $0.85_{\pm0.01}$ | $0.83_{\pm0.02}$ |
| **TD($\lambda$)-UCB** | $\mathbf{0.94_{\pm0.00}}$ | $\mathbf{0.93_{\pm0.00}}$ | $\mathbf{0.93_{\pm0.00}}$ | $\mathbf{0.92_{\pm0.00}}$ | $\mathbf{0.91_{\pm0.01}}$ | $\mathbf{0.91_{\pm0.01}}$ | $\mathbf{0.90_{\pm0.01}}$ | $\mathbf{0.89_{\pm0.02}}$ | $\mathbf{0.88_{\pm0.02}}$ |

| | **SplitMNIST-Hard** | | | | **SplitNotMNIST-Hard** | | | |
|---|---|---|---|---|---|---|---|---|
| | t = 2 | t = 3 | t = 4 | t = 5 | t = 2 | t = 3 | t = 4 | t = 5 |
| VCL | $0.87_{\pm0.02}$ | $0.66_{\pm0.04}$ | $0.82_{\pm0.03}$ | $0.64_{\pm0.11}$ | $0.69_{\pm0.04}$ | $0.63_{\pm0.03}$ | $0.60_{\pm0.00}$ | $0.51_{\pm0.06}$ |
| **TD($\lambda$)-VCL** | $\mathbf{0.98_{\pm0.01}}$ | $\mathbf{0.79_{\pm0.08}}$ | $\mathbf{0.88_{\pm0.04}}$ | $\mathbf{0.67_{\pm0.04}}$ | $\mathbf{0.74_{\pm0.02}}$ | $\mathbf{0.73_{\pm0.03}}$ | $\mathbf{0.69_{\pm0.03}}$ | $\mathbf{0.58_{\pm0.09}}$ |
| UCL | $0.88_{\pm0.04}$ | $0.68_{\pm0.03}$ | $0.83_{\pm0.03}$ | $0.66_{\pm0.06}$ | $0.71_{\pm0.01}$ | $0.63_{\pm0.04}$ | $0.61_{\pm0.00}$ | $\mathbf{0.52_{\pm0.04}}$ |
| **TD($\lambda$)-UCL** | $\mathbf{0.97_{\pm0.01}}$ | $\mathbf{0.85_{\pm0.06}}$ | $\mathbf{0.90_{\pm0.02}}$ | $\mathbf{0.70_{\pm0.04}}$ | $\mathbf{0.72_{\pm0.03}}$ | $\mathbf{0.71_{\pm0.06}}$ | $\mathbf{0.63_{\pm0.02}}$ | $0.51_{\pm0.06}$ |
| UCB | $0.85_{\pm0.16}$ | $0.79_{\pm0.12}$ | $0.83_{\pm0.06}$ | $0.75_{\pm0.10}$ | $0.70_{\pm0.08}$ | $0.63_{\pm0.06}$ | $0.61_{\pm0.01}$ | $0.61_{\pm0.05}$ |
| **TD($\lambda$)-UCB** | $\mathbf{0.93_{\pm0.02}}$ | $\mathbf{0.89_{\pm0.03}}$ | $\mathbf{0.87_{\pm0.03}}$ | $\mathbf{0.80_{\pm0.03}}$ | $\mathbf{0.72_{\pm0.01}}$ | $\mathbf{0.72_{\pm0.01}}$ | $\mathbf{0.70_{\pm0.02}}$ | $\mathbf{0.63_{\pm0.03}}$ |

**CIFAR100-10**

| | t = 2 | t = 3 | t = 4 | t = 5 | t = 6 | t = 7 | t = 8 | t = 9 | t = 10 |
|---|---|---|---|---|---|---|---|---|---|
| VCL | $0.64_{\pm0.02}$ | $0.63_{\pm0.03}$ | $0.63_{\pm0.02}$ | $0.60_{\pm0.02}$ | $0.60_{\pm0.02}$ | $0.60_{\pm0.03}$ | $0.61_{\pm0.05}$ | $0.65_{\pm0.02}$ | $0.66_{\pm0.01}$ |
| **TD($\lambda$)-VCL** | $\mathbf{0.66_{\pm0.02}}$ | $\mathbf{0.67_{\pm0.02}}$ | $\mathbf{0.66_{\pm0.04}}$ | $\mathbf{0.66_{\pm0.01}}$ | $\mathbf{0.66_{\pm0.02}}$ | $\mathbf{0.66_{\pm0.01}}$ | $\mathbf{0.67_{\pm0.01}}$ | $\mathbf{0.69_{\pm0.02}}$ | $\mathbf{0.71_{\pm0.01}}$ |
| UCL | $0.65_{\pm0.03}$ | $0.66_{\pm0.07}$ | $0.64_{\pm0.05}$ | $0.62_{\pm0.04}$ | $0.60_{\pm0.05}$ | $0.60_{\pm0.04}$ | $0.58_{\pm0.02}$ | $0.61_{\pm0.02}$ | $0.62_{\pm0.02}$ |
| **TD($\lambda$)-UCL** | $\mathbf{0.68_{\pm0.02}}$ | $\mathbf{0.67_{\pm0.02}}$ | $\mathbf{0.64_{\pm0.01}}$ | $\mathbf{0.70_{\pm0.04}}$ | $\mathbf{0.70_{\pm0.02}}$ | $\mathbf{0.68_{\pm0.03}}$ | $\mathbf{0.66_{\pm0.03}}$ | $\mathbf{0.65_{\pm0.06}}$ | $\mathbf{0.67_{\pm0.03}}$ |
| UCB | $\mathbf{0.65_{\pm0.01}}$ | $\mathbf{0.65_{\pm0.02}}$ | $\mathbf{0.66_{\pm0.02}}$ | $0.66_{\pm0.03}$ | $0.66_{\pm0.03}$ | $0.66_{\pm0.01}$ | $0.65_{\pm0.01}$ | $0.64_{\pm0.01}$ | $0.66_{\pm0.01}$ |
| **TD($\lambda$)-UCB** | $\mathbf{0.64_{\pm0.02}}$ | $\mathbf{0.65_{\pm0.02}}$ | $\mathbf{0.66_{\pm0.01}}$ | $\mathbf{0.67_{\pm0.01}}$ | $\mathbf{0.67_{\pm0.01}}$ | $\mathbf{0.68_{\pm0.01}}$ | $\mathbf{0.68_{\pm0.01}}$ | $\mathbf{0.68_{\pm0.02}}$ | $\mathbf{0.70_{\pm0.01}}$ |

**TinyImagenet-10**

| | t = 2 | t = 3 | t = 4 | t = 5 | t = 6 | t = 7 | t = 8 | t = 9 | t = 10 |
|---|---|---|---|---|---|---|---|---|---|
| VCL | $0.53_{\pm0.06}$ | $0.50_{\pm0.02}$ | $0.51_{\pm0.03}$ | $0.52_{\pm0.02}$ | $0.51_{\pm0.03}$ | $0.49_{\pm0.01}$ | $0.51_{\pm0.02}$ | $0.51_{\pm0.02}$ | $0.51_{\pm0.02}$ |
| **TD($\lambda$)-VCL** | $\mathbf{0.57_{\pm0.03}}$ | $\mathbf{0.55_{\pm0.02}}$ | $\mathbf{0.56_{\pm0.02}}$ | $\mathbf{0.56_{\pm0.01}}$ | $\mathbf{0.55_{\pm0.03}}$ | $\mathbf{0.55_{\pm0.03}}$ | $\mathbf{0.56_{\pm0.02}}$ | $\mathbf{0.57_{\pm0.02}}$ | $\mathbf{0.56_{\pm0.02}}$ |
| UCL | $0.55_{\pm0.02}$ | $0.52_{\pm0.03}$ | $0.52_{\pm0.03}$ | $0.52_{\pm0.02}$ | $0.51_{\pm0.02}$ | $0.50_{\pm0.02}$ | $0.52_{\pm0.01}$ | $0.52_{\pm0.01}$ | $0.50_{\pm0.03}$ |
| **TD($\lambda$)-UCL** | $\mathbf{0.55_{\pm0.03}}$ | $\mathbf{0.53_{\pm0.01}}$ | $\mathbf{0.54_{\pm0.01}}$ | $\mathbf{0.55_{\pm0.01}}$ | $\mathbf{0.54_{\pm0.01}}$ | $\mathbf{0.54_{\pm0.01}}$ | $\mathbf{0.55_{\pm0.01}}$ | $\mathbf{0.56_{\pm0.01}}$ | $\mathbf{0.56_{\pm0.01}}$ |
| UCB | $0.52_{\pm0.06}$ | $0.51_{\pm0.04}$ | $0.51_{\pm0.02}$ | $0.50_{\pm0.02}$ | $0.48_{\pm0.04}$ | $0.46_{\pm0.01}$ | $0.45_{\pm0.02}$ | $0.44_{\pm0.03}$ | $0.42_{\pm0.03}$ |
| **TD($\lambda$)-UCB** | $\mathbf{0.54_{\pm0.04}}$ | $\mathbf{0.54_{\pm0.01}}$ | $\mathbf{0.52_{\pm0.01}}$ | $\mathbf{0.52_{\pm0.02}}$ | $\mathbf{0.51_{\pm0.02}}$ | $\mathbf{0.50_{\pm0.02}}$ | $\mathbf{0.50_{\pm0.03}}$ | $\mathbf{0.49_{\pm0.02}}$ | $\mathbf{0.47_{\pm0.02}}$ |

# N   Does TD-VCL Assume Knowledge of Task Boundaries?

In this Section, we argue that the TD-VCL objective (and VCL objectives in general) does not require knowledge of task boundaries, and we provide theoretical and empirical evidence for that. The theoretical argument comes from the principle that the *Bayesian framework is self-consistent*: given a stream of data, the final posterior distribution should be the same regardless of how many Bayesian updates are executed.

Based on that, the key thing is to realize that the number of updates does not need to be equal to the number of tasks. Mathematically, suppose we have a stream of $T$ tasks (represented by $t$). At a particular update $k$, we may consider a Bayesian update that includes data from multiple ($m$) sequential tasks (e.g., from $t_a$ to $t_{a+m}$):

$$\mathcal{D}_k = \bigcup_{j=0}^{m} \mathcal{D}_j. \tag{24}$$

Crucially, this does not impose any assumptions on boundaries. Rather, once we decide where to start and end the data stream for the Bayesian update, there could be potentially many tasks included. Under the same assumptions stated in Section 3, we have that:

$$p(\mathcal{D}_k \mid \theta) = \prod_{j=0}^{m} p(\mathcal{D}_j \mid \theta). \tag{25}$$

And, the recursive relationship (Equation 1) also follows:

$$p(\theta \mid \mathcal{D}_{1:k}) \propto p(\theta \mid \mathcal{D}_{1:k-1}) \prod_{j=0}^{m} p(\mathcal{D}_k \mid \theta). \tag{26}$$

Finally, following the same variational objective and ELBO derivation, we arrive at

$$\mathcal{L}^k(\theta) = \mathbb{E}_{\theta \sim q_k(\theta)} \left[ \sum_{j=0}^{m} \log p(\mathcal{D}_k \mid \theta) \right] - \mathcal{D}_{\mathrm{KL}}(q_k(\theta) \,\|\, q_{k-1}(\theta)). \tag{27}$$

Therefore, the objective itself does not discriminate or require task boundaries. TD-VCL will estimate the likelihood terms for multiple terms simultaneously, which is something already done while replaying past tasks.

**Empirical Evidence.** We highlight that most benchmarks – including the ones presented in this work – isolate tasks, which makes it convenient to consider one Bayesian update per task. To provide further practical evidence that the method does not require knowledge of boundaries, we present another benchmark called **StreamingPermutedMNIST-Hard**. This benchmark does not provide any boundary between tasks. From the full data stream of $T$ tasks, we create sequential streams of data where boundaries are placed randomly and provide them to the methods for continual learning. We execute an evaluation after the complete data stream, considering held-out splits composed of all tasks. We report the average accuracy across them, equivalently to the $t = 10$ column in Tables 2 and 3. Table 9 shows the empirical results over 10 seeds. We observe no negative impact in the VCL/TD-VCL methods in comparison with PermutedMNIST-Hard. In fact, some methods improved performance, because we are likely replaying the same task into different chunks, alleviating the catastrophic forgetting challenge. The proposed objectives still outperform all other methods.

Table 9: **StreamingPermutedMNIST-Hard results**. We observe no negative impact in the TD-VCL methods in comparison with PermutedMNIST-Hard, suggesting that these methods do not require knowledge of task boundaries.

| StreamingPermutedMNIST-Hard | |
|---|---|
| **Method** | $t = 10$ |
| Online MLE | $0.54_{\pm 0.09}$ |
| Batch MLE | $0.64_{\pm 0.09}$ |
| VCL | $0.82_{\pm 0.05}$ |
| VCL CoreSet | $0.85_{\pm 0.04}$ |
| **N-Step TD-VCL** | $\mathbf{0.89_{\pm 0.02}}$ |
| **TD($\lambda$)-VCL** | $\mathbf{0.89_{\pm 0.02}}$ |

# O   Further Questions

This Appendix presents additional clarification questions aimed at improving the understanding of the proposed method and experiments. These questions were raised during the peer-review process, and we refer to the OpenReview page for the full discussion.

## O.1   What is the computational cost associated with TD-VCL?

We analyze the computational cost from three aspects: training, inference, and hyperparameter search.

**Training Cost.** The training cost arises from the computation of Equations 4 and 5, which depend on two components: (I) Monte Carlo estimation of the likelihood term, and (II) the KL regularization term. (I) corresponds to the standard cross-entropy loss for classification, averaged over samples of $\theta$ from the variational distribution. *In practice, we approximate this average with a single sample*, so the cost is equivalent to standard classification under the MLE objective. (II) is the KL regularization term, which can be computed in closed form. This computation is lightweight since it does not involve data or forward passes through the network. Overall, the training costs of VCL and TD-VCL are nearly identical, as the main bottleneck lies in (I), which is similar in both methods. Additionally, we employ early stopping, which reduces the number of training epochs and thereby lowers computational requirements compared to prior implementations.

**Inference Cost.** Bayesian inference is approximated by the posterior predictive distribution $p(y^* \mid \mathbf{x}^*, \mathcal{D}_{1:t}) = \mathbb{E}_{q_t(\theta)}\big[p(y^* \mid \theta, \mathbf{x}^*, \mathcal{D}_{1:t})\big]$, which we estimate via Monte Carlo sampling. The computational cost of this step depends on the number of parameter samples $\theta$. In our experiments, *we use a single sample* to ensure computational fairness across all baselines. With this choice, inference reduces to a single forward pass through the network, identical to a standard classifier. Using more samples increase the cost proportionally but also improve predictive performance.

**Hyperparameter Search Cost.** The cost of hyperparameter search can be managed by constraining the computational budget. As described in Appendix H, we restrict all methods (including baselines) to a budget of at most one GPU-day. Our method introduces two additional hyperparameters beyond those of VCL ($n$ and $\lambda$). Results in Appendix L show moderate to good robustness to these choices, suggesting that the search cost could be further reduced if necessary.

## O.2   What is the memory cost of TD-VCL?

We analyze memory usage in terms of maintaining the replay buffer and storing the previous posteriors.

**Replay Buffer Cost.** The buffer has memory complexity $\mathcal{O}(n)$, where $n$ is the $n$-Step hyperparameter. This cost is no greater than that of the core sets used in VCL CoreSet. In our benchmarks, the replay buffer is intentionally limited to at most 200 data points from previously observed tasks, which is negligible compared to the 60,000 data points in the current task. As a result, the replay buffer does not represent a major bottleneck.

**Posterior Storage Cost.** The storage of posteriors also has memory complexity $\mathcal{O}(n)$. For smaller networks (such as those used in the MNIST and NotMNIST benchmarks), the memory usage is comparable to that of the replay buffer: for example, storing 200 MNIST data points requires about 0.60 MB, while storing the posterior requires about 0.68 MB (assuming float32 precision). Naturally, the memory required for posteriors increases with larger and deeper networks.

## O.3   Is maintaining previous posteriors a major bottleneck? Can we optimize this cost?

As noted in the Limitations (Section 6), TD-VCL may increase memory requirements. However, this is not necessarily a major limitation, and we also discuss strategies to reduce memory usage.

**Memory Complexity is a function of $n$, which the user controls**. The buffer size and number of posteriors are defined by $n$. If memory is a bottleneck, one can control $n$ to satisfy memory constraints. *Crucially, $n$ is not always equal to the number of tasks.* Our robustness analysis (Appendix L) shows that performance increases monotonically with $n$ up to a level where it may saturate. Therefore, any $n > 1$ should be better than vanilla VCL, and, if performance is expected to saturate, one can also set

$n$ to be much lower than the number of tasks. The employed hyperparameters (Appendix H) suggest that we can usually assume a value of $n$ that is lower than the number of tasks.

**Assume a memory-efficient variational family** $\mathcal{Q}$. Since memory may be a challenge for large Bayesian networks, there are alternative architectures, such as last-layer variational methods or bayesian LoRA adapters [59, 12, 52], which approximates the posterior distribution in a fixed number of parameters. These methods drastically reduce the required memory at the cost of expressiveness of the variational family.

**Store previous posteriors in cheaper memory alternatives**. Since TD-VCL does not use previous posteriors for inference but only for computing the KL regularization, they do not need to occupy GPU memory. In fact, the regularization term could be computed asynchronously on CPU (or even with an external computer) while the GPU is used to generate predictions and estimate the likelihood terms. While implementation is more involved, it allows the use of both CPU/GPU and avoids having previous posteriors in GPU memory, which is usually the bottleneck.

**Estimate TD objective with fewer posteriors but covering older timesteps**. A corollary of Proposition 4.4 is that we can represent the learning target as **any** combination of $n$-step TD targets. This means that we may store posteriors at every $m$ steps, instead of every step. In this case, given $T$ tasks, we only store $T/m$ posteriors. Naturally, this leads to a different way of estimating the learning objective, but ensures that it is covering older tasks to prevent catastrophic forgetting.

Lastly, we highlight that there are realistic Continual Learning settings where storing posteriors is not a major bottleneck. For instance, when *continually learning on embedded systems with access to a cloud storage*. These embedded systems (mobile phones, wearables) usually present limited storage/GPU memory onboard. Some problems require on-the-fly model adaptation and, for privacy-related reasons, the data must be kept on the device for a limited time. Nonetheless, we may upload model snapshots to the cloud without problems (sometimes this upload is required to conduct quality evaluation or audits). A concrete example is an on-device speech recognition model on a smartwatch adapting to a user's voice. We believe our TD-VCL objective is well-suited for this problem setting.

## O.4 When should one use $n$-Step TD-VCL or TD($\lambda$)-VCL?

TD($\lambda$)-VCL is a generalization of $n$-Step TD-VCL. As we presented in Appendix E, TD($\lambda$)-VCL forms a spectrum of CL algorithms, and we recover $n$-Step TD-VCL when $\lambda \to 1$. Therefore, the "choice" depends on $\lambda$, which controls how much the learning objective should prioritize recent posteriors. If one believes that the most recent posterior retains the knowledge of previous tasks, then a higher $\lambda$ should work better. Otherwise, one should use lower values as past estimates contain information that has not propagated over the recursive updates. In practice, it depends on the continual learning problem and the potential transfer/interference among tasks. The recommendation is to start from TD($\lambda$)-VCL and tune the $\lambda$ hyperparameter.

## O.5   What is the impact of Early Stopping in the presented methods?

We perform an ablation study to evaluate the impact of Early Stopping, with results reported in Table 10. We find that removing Early Stopping does not significantly impact the performance of TD-VCL, while VCL shows a slight degradation. The methods most negatively affected are the MLE baselines. This result is expected, since these models lack any form of regularization, and training without Early Stopping leads to overfitting. Notably, experiments without Early Stopping required approximately 5 to 10 times more training time. As discussed in the main paper, Early Stopping substantially reduces computational cost.

Table 10: **Results without Early Stopping.** We observe that TD-VCL maintains strong performance even without early stopping, while VCL shows slight degradation and MLE baselines suffer from overfitting. Training without early stopping also took 5–10× longer.

| | PermutedMNIST-Hard | | | | | | | | |
| | t = 2 | t = 3 | t = 4 | t = 5 | t = 6 | t = 7 | t = 8 | t = 9 | t = 10 |
|---|---|---|---|---|---|---|---|---|---|
| Online MLE | $0.75_{\pm0.04}$ | $0.58_{\pm0.07}$ | $0.55_{\pm0.07}$ | $0.44_{\pm0.05}$ | $0.42_{\pm0.05}$ | $0.36_{\pm0.04}$ | $0.34_{\pm0.05}$ | $0.31_{\pm0.04}$ | $0.30_{\pm0.04}$ |
| Batch MLE | $0.94_{\pm0.01}$ | $0.89_{\pm0.02}$ | $0.82_{\pm0.04}$ | $0.69_{\pm0.03}$ | $0.64_{\pm0.04}$ | $0.55_{\pm0.03}$ | $0.52_{\pm0.03}$ | $0.46_{\pm0.04}$ | $0.45_{\pm0.03}$ |
| VCL | $0.96_{\pm0.01}$ | $0.94_{\pm0.01}$ | $0.92_{\pm0.01}$ | $0.89_{\pm0.04}$ | $0.86_{\pm0.04}$ | $0.84_{\pm0.04}$ | $0.80_{\pm0.09}$ | $0.78_{\pm0.11}$ | $0.71_{\pm0.15}$ |
| VCL CoreSet | $0.96_{\pm0.00}$ | $0.95_{\pm0.01}$ | $0.94_{\pm0.01}$ | $0.92_{\pm0.02}$ | $0.90_{\pm0.02}$ | $0.88_{\pm0.02}$ | $0.85_{\pm0.02}$ | $0.83_{\pm0.03}$ | $0.79_{\pm0.09}$ |
| **N-Step TD-VCL** | $\mathbf{0.95_{\pm0.00}}$ | $\mathbf{0.95_{\pm0.00}}$ | $\mathbf{0.94_{\pm0.00}}$ | $\mathbf{0.93_{\pm0.00}}$ | $\mathbf{0.92_{\pm0.01}}$ | $\mathbf{0.92_{\pm0.01}}$ | $0.89_{\pm0.01}$ | $0.88_{\pm0.02}$ | $0.85_{\pm0.04}$ |
| **TD($\lambda$)-VCL** | $\mathbf{0.97_{\pm0.00}}$ | $\mathbf{0.97_{\pm0.00}}$ | $\mathbf{0.96_{\pm0.01}}$ | $\mathbf{0.95_{\pm0.00}}$ | $\mathbf{0.94_{\pm0.01}}$ | $\mathbf{0.93_{\pm0.01}}$ | $\mathbf{0.91_{\pm0.01}}$ | $\mathbf{0.91_{\pm0.02}}$ | $\mathbf{0.89_{\pm0.02}}$ |

