# OpenReview forum: "Temporal-Difference Variational Continual Learning"
_NeurIPS.cc/2025/Conference — NeurIPS 2025 poster_

### Official Review · Reviewer_xMRf · 2025-06-29

**Clarity:** 3
**Significance:** 3
**Originality:** 3
**Rating:** 5
**Confidence:** 4

**Summary:**

The authors propose an extension to the Variational Continual Learning (VCL) framework, which performs online variational inference by using the posterior from the previous task as the prior for the current one. To address the accumulation of approximation errors across tasks, they introduce new training objectives that combine likelihoods and regularization terms from both current and past tasks using weighted sums. Notably, they show that the original VCL objective is a special case of their formulation for a particular choice of weights.

**Questions:**

* Can the author include the upper bound on the average accuracy achieved by training on all tasks up to time t?
* Early stopping is used in the experiments. Could the authors include an ablation study to assess its effect on performance and stability?
*  The buffer size, which is critical for understanding the experimental setup, is only mentioned in the appendix. Could the authors move this information into the main paper?
* Would it be possible to extend this method to task-agnostic settings, or are there theoretical or empirical reasons preventing this?

I am open to increasing my overall score if the authors adequately address my concerns, particularly regarding the baseline comparisons, memory analysis, and assumptions about task boundaries.

**Ethical Concerns:**

["NO or VERY MINOR ethics concerns only"]

**Final Justification:**

The authors have fully addressed my main concerns.

**Limitations:**

* The proposed method is task-aware and requires explicit knowledge of task boundaries to adjust the loss (e.g., update the regularization term and buffer). Consequently, it is not applicable in task-agnostic or non-stationary scenarios where changes in data distribution occur without clear segmentation.
* As with many Bayesian continual learning methods, the evaluation is limited to fully connected networks and AlexNet, which does not incorporate modern architectural features such as Batch Normalization. While this is not unique to this work, it reflects a broader limitation of Bayesian deep learning methods that deserves acknowledgment.

**Quality:**

3

**Strengths And Weaknesses:**

***Strengths:***

* The authors leverage the Bayesian framework to introduce a principled method that incorporates information from both current and past tasks, while addressing the accumulation of approximation errors inherent in online variational inference.

* The experimental results show that the proposed method improves upon VCL (with/without CoreSet) across several datasets under the reported experimental settings.

***Weaknesses:***

*	The preliminaries lack important context. In particular, the authors do not explicitly state a key assumption of their method: that task boundaries are known. This assumption is critical for applying the approach and should be clearly stated in the introduction and discussed in the related work section.
Relevant task-agnostic Bayesian approaches include:
1.	Bonnet et al., Bayesian Continual Learning and Forgetting in Neural Networks, arXiv 2025
2.	Zeno et al., Task-Agnostic Continual Learning Using Online Variational Bayes with Fixed-Point Updates, Neural Computation 2021

* Since the method is task-aware, a stronger baseline for online maximum likelihood estimation (MLE) should be included. For instance, Mirzadeh et al. (2020) report over 80% average accuracy after 20 tasks, significantly higher than the ~40% reported in this paper after only 10 tasks, by using dropout and carefully tuned hyperparameters. A fair comparison would require similarly optimized training setups.
Reference: Mirzadeh et al., “Understanding the Role of Training Regimes in Continual Learning,” NeurIPS 2020.
* As the method relies on storing past posteriors (i.e., the mean and variance of each weight), the authors should analyze the memory requirements of this representation and compare it with the buffer size used in each experiment (typically 40–200 examples). In addition, a comparison with rehearsal-based methods should be included—for example:
Buzzega et al., “Dark Experience for General Continual Learning: A Strong, Simple Baseline,” NeurIPS 2020.

---

> ### Author Rebuttal · Authors · 2025-07-30
>
> Thank you for your review! We appreciate that you found our work **principled**, with **good quality and originality.**
>
> We aim to address your concerns below:
>
> **Q12** Does TD-VCL assume knowledge of task boundaries?
>
> **A12** We kindly refer the reviewer to our answer **A8** (under reviewer **aVHF**'s review). In summary, we clarify that our method does NOT assume task boundaries and provide theoretical and empirical evidence.
>
> **Q13** Relevant task-agnostic Bayesian approaches (Bonnet et. al., Zeno et. al.)
>
> **A13** Thank you for raising this literature. We will discuss these works here and add such discussion to the Related Work section.
>
> The work of Bonnet et. al. [1] (contemporaneous to ours) proposes a forgetting mechanism for older tasks. Similarly to our work, they start from the VCL objective (Eq. 5 in [1]). In contrast, they add a positive likelihood term to the loss in order to forget old tasks. Since our work focuses on **preventing** catastrophic forgetting, this mechanism would actually hinder performance. In summary, these methods concern different problem settings.
>
> The work of Zeno et. al. [2] derives an analytical solution to the optimization problem via fixed-point matrix equations. Similarly to our work, they also start from the VCL objective (Eq. 6 in [2]). In contrast, we opted for a deep learning approach and optimized via stochastic gradient descent. We believe that a fixed-point matrix solution might not scale well with the number of parameters. In any case, the TD objectives may also be applied here. Our work proposes a new optimization objective, but does not impose how to solve it.
>
> We also highlight that both methods are indeed task-agnostic due to the same reasons as our TD-VCL objective (described in **A8**).
>
> **Q14** Stronger MLE baseline from Mirzadeh et al [3].
>
> **A14** Thank you for highlighting this work. We analyzed this paper and experiments thoroughly, as the baseline results seemed surprisingly strong.
>
> We carefully looked at their provided code [4] and executed it exactly as documented for PermutedMNIST. We found the following results for their method "Stable SGD":
>
> | t = 2   | t = 4   | t = 6   | t = 8   | t = 10  | t = 20  |
> |--------|--------|--------|--------|--------|--------|
> | 0.95±0.02 | 0.88±0.04 | 0.79±0.03 | 0.72±0.03 | 0.63±0.03 | 0.41±0.03 |
>
> Our first point is that **the code does not reproduce the results in the paper**, which claimed an 80% average after 20 tasks. With further investigation, we found that other researchers tried to reproduce the same results without success ([4], github issue 5). The authors provided responses regarding other benchmarks in the paper, but never for PermutedMNIST. Interestingly, one of the original reviewers of this paper also could not reproduce the results ([5], see Reviewer 3 "Weaknesses"), and the authors only responded by considering the other benchmarks, but not PermutedMNIST ([6], see "R3-Reproducibility"). Therefore, it is unclear if these results are really reproducible.
>
> We also question their experimental design. It involves a strong search in very sensitive hyperparameters (learning rate and decay, batch size, dropout), using the test set as a validation objective *exhaustively*. This process may lead to overoptimizing the metric for that test set, which we believe is not a good practice for ML research. In our case, we adopted the analogous hyperparameters consistently with prior work to establish a fair comparison.
>
> Lastly, if we consider this tuning procedure acceptable, we argue that all methods in our paper would benefit from it. Indeed, Mirzadeh et. al. themselves [3] show that this tuning procedure improves all baselines (Table 4 in [3]). Therefore, we do not expect this to change our conclusion that the TD-VCL objective helps address catastrophic forgetting.
>
> **Q15** Memory Cost Analysis
>
> **A15** We kindly refer the reviewer to our answer **A5** under reviewer **LVzR**, where we describe the memory requirements and discuss it.
>
> **Q16** Can the author include the upper bound on the average accuracy achieved by training on all tasks up to time t?
>
> **A16** We refer the reviewer to Figure 4 (left) in our paper, specifically for the curve where T = 10 and B = 60,000, which effectively means training in all past tasks with the full available data. The average accuracy is ~97%.
>
> **Q17** Early stopping is used in the experiments. Could the authors include an ablation study to assess its effect on performance and stability?
>
> **A17** Thank you for the suggestion. We report here the results without early stopping. We observe that the absence of Early Stopping does not affect the performance of TD-VCL. VCL seems to have a slight degradation in performance. The only ones that really struggle are the MLE baselines. This makes sense as they do not have any form of regularization, and the absence of early stopping leads to overfitting.
>
> More interestingly, we highlight that running experiments without Early Stopping took around 5 to 10 times longer. As we stated in the paper, Early Stopping considerably reduces computation time.
>
>
> | Method Name     | t = 2       | t = 3       | t = 4       | t = 5       | t = 6       | t = 7       | t = 8       | t = 9       | t = 10       |
> |-----------------|-------------|-------------|-------------|-------------|-------------|-------------|-------------|-------------|--------------|
> | Online MLE      | 0.75 ± 0.04 | 0.58 ± 0.07 | 0.55 ± 0.07 | 0.44 ± 0.05 | 0.42 ± 0.05 | 0.36 ± 0.04 | 0.34 ± 0.05 | 0.31 ± 0.04 | 0.30 ± 0.04  |
> | Batch MLE       | 0.94 ± 0.01 | 0.89 ± 0.02 | 0.82 ± 0.04 | 0.69 ± 0.03 | 0.64 ± 0.04 | 0.55 ± 0.03 | 0.52 ± 0.03 | 0.46 ± 0.04 | 0.45 ± 0.03  |
> | VCL             | 0.96 ± 0.01 | 0.94 ± 0.01 | 0.92 ± 0.01 | 0.89 ± 0.04 | 0.86 ± 0.04 | 0.84 ± 0.04 | 0.80 ± 0.09 | 0.78 ± 0.11 | 0.71 ± 0.15  |
> | VCL CoreSet     | 0.96 ± 0.00 | 0.95 ± 0.01 | 0.94 ± 0.01 | 0.92 ± 0.02 | 0.90 ± 0.02 | 0.88 ± 0.02 | 0.85 ± 0.02 | 0.83 ± 0.03 | 0.79 ± 0.09  |
> | N-Step TD-VCL   | 0.95 ± 0.00 | 0.95 ± 0.00 | 0.94 ± 0.00 | 0.93 ± 0.00 | 0.92 ± 0.01 | 0.92 ± 0.01 | 0.89 ± 0.01 | 0.88 ± 0.02 | 0.85 ± 0.04  |
> | TD(λ)-VCL       | 0.97 ± 0.00 | 0.97 ± 0.00 | 0.96 ± 0.01 | 0.95 ± 0.00 | 0.94 ± 0.01 | 0.93 ± 0.01 | 0.91 ± 0.01 | 0.91 ± 0.02 | 0.89 ± 0.02  |
>
>
> **Q18**  The buffer size, which is critical for understanding the experimental setup, is only mentioned in the appendix. Could the authors move this information into the main paper?
>
> **A18** Thank you for the suggestion. We added this information in the Benchmarks paragraph of Section 5.
>
> **Q19** Would it be possible to extend this method to task-agnostic settings, or are there theoretical or empirical reasons preventing this? and
>
> Limitation: The proposed method is task-aware and requires explicit knowledge of task boundaries to adjust the loss (e.g., update the regularization term and buffer). Consequently, it is not applicable in task-agnostic or non-stationary scenarios where changes in data distribution occur without clear segmentation.
>
> **A19** We hope our answer **A8** under reviewer **aVHF**'s response clarifies these points.
>
> **Q20** Limitation: As with many Bayesian continual learning methods, the evaluation is limited to fully connected networks and AlexNet, which does not incorporate modern architectural features such as Batch Normalization. While this is not unique to this work, it reflects a broader limitation of Bayesian deep learning methods that deserves acknowledgment.
>
> **A20** Our choice of deep network architectures indeed follows prior work in the area [7, 8, 9]. Nonetheless, we highlight that stabilizing variational architectures is an active area of research. Indeed, there are works that bring Bayesian ResNets [10] (which incorporates batch norm) and Bayesian Transformers [11] -- perhaps not yet validated in CL settings. We also believe that validating building blocks like MLPs already unlocks applications in deeper networks via last-layer Bayesian methods [12] or Bayesian adapters [13]. In summary, we do agree that this area is still in development and presents limitations, but we also believe that the employed architectures are significant and provide solid evidence to support our claims related to the TD-VCL objectives.
>
> References
>
> [1] Bonnet et al., Bayesian Continual Learning and Forgetting in Neural Networks, 2025.
>
> [2] Zeno et al., Task-Agnostic Continual Learning Using Online Variational Bayes with Fixed-Point Updates. Neural Computation, 2021
>
> [3] Mirzadeh et al. Understanding the Role of Training Regimes in Continual Learning. NeurIPS, 2020.
>
> [4] Code for "Understanding the Role of Training Regimes in Continual Learning". Available in "github: imirzadeh/stable-continual-learning". Accessed in July 2025.
>
> [5] NeurIPS official review for "Understanding the Role of Training Regimes in Continual Learning". Available in "proceedings.neurips.cc: paper_files/paper/2020/file/518a38cc9a0173d0b2dc088166981cf8-Review.html". Accessed in July 2025.
>
> [6] NeurIPS Offical Author Feedback for "Understanding the Role of Training Regimes in Continual Learning". Available in "proceedings.neurips.cc :paper_files/paper/2020/file/518a38cc9a0173d0b2dc088166981cf8-AuthorFeedback.pdf". Accessed in July 2025.
>
> [7] Nguyen et. al. Variational Continual Learning. ICLR, 2018.
>
> [8] Ahn et. al. Uncertainty-based Continual Learning with
>  Adaptive Regularization. NeurIPS, 2019.
>
> [9] Ebrahimi et. al. Uncertainty-Guided Continual Learning with Bayesian Neural Networks. ICLR, 2020.
>
> [10] Krishnan et. al. Efficient Priors for Scalable Variational Inference in Bayesian Deep Neural Networks. ICCVW, 2019.
>
> [11] Sankararaman et al. BayesFormer: Transformer with Uncertainty Estimation, 2022.
>
> [12] Melo et. al. Deep Bayesian Active Learning for Preference Modeling in Large Language Models. NeurIPS, 2024.
>
> [13] Yang et. al. Bayesian Low-rank Adaptation for Large Language Models. ICLR, 2024.

---

> ### Comment · Reviewer_xMRf · 2025-08-03
>
> I thank the authors for their thoughtful responses to my questions and concerns.
>
> ***Comments and Concerns:***
> * I agree with the authors that the online variational Bayes objective does not theoretically require knowledge of task boundaries. In continual learning settings, once tasks have changed, we no longer have access to data from previous tasks. However, in the StreamingPermutedMNIST-Hard experiment, the randomly sampled data chunks appear to contain data from multiple tasks (as acknowledged by the authors in the sentence: “because we are likely replaying the same task into different chunks”). This seems to violate the assumptions of the continual learning setting.
> * I also thank the authors for carefully analyzing the Stable SGD method and improving the baseline performance after 10 tasks by approximately 20%, which is a meaningful improvement. While the tuning procedure benefits all baselines, the improvement of vanilla SGD (with regularization) is particularly noteworthy, as it reflects the benchmark's challenge and relevance.

---

> > ### Author Response · Authors · 2025-08-05
> >
> > Dear reviewer xMRf,
> >
> > Thank you for reading our rebuttal and providing further comments.
> >
> > **Re StableSGD**: Thank you for your comment. We agree with it, especially regarding the benchmark's challenge and relevance. We are happy to incorporate Stable SGD results in the PermutedMNIST-Hard as another MLE-based baseline.
> >
> > **Re StreamingPermutedMNIST:** Thank you for raising this point. We would like to make a clarification on our data chunking approach (apologies for not being clear in the first message): We chunk **sequential** streams of data, where the **boundaries are placed randomly**. We provide a graphical example below, with 4 tasks and 4 update steps. ●₁ represents a data point of task 1, and the graphic illustrates the streaming of points over time (from left to right).
> >
> > **Standard PermutedMNIST-Hard:** Task boundaries coincide with update boundaries.
> >
> > $$
> > \underbrace{
> >   \bullet_1\ \bullet_1\ \bullet_1\ \bullet_1\ \bullet_1\ \bullet_1
> > }\_{\text{Update 1}}
> > \hspace{5pt} | \hspace{5pt}
> > \underbrace{
> >   \bullet_2\ \bullet_2\ \bullet_2\ \bullet_2\ \bullet_2\ \bullet_2
> > }\_{\text{Update 2}}
> > \hspace{5pt} | \hspace{5pt}
> > \underbrace{
> >   \bullet_3\ \bullet_3\ \bullet_3\ \bullet_3\ \bullet_3\ \bullet_3
> > }\_{\text{Update 3}}
> > \hspace{5pt} | \hspace{5pt}
> > \underbrace{
> >   \bullet_4\ \bullet_4\ \bullet_4\ \bullet_4\ \bullet_4\ \bullet_4
> > }\_{\text{Update 4}}
> > $$
> >
> >
> > **StreamingPermutedMNIST-Hard:** Task boundaries are unknown; therefore, update boundaries are set randomly.
> >
> > $$
> > \underbrace{
> >   \bullet_1\ \bullet_1\ \bullet_1\ \bullet_1\ \bullet_1\ \bullet_1\ \bullet_2\ \bullet_2
> > }\_{\text{Update 1}}
> > \hspace{5pt} | \hspace{5pt}
> > \underbrace{
> >   \bullet_2\ \bullet_2\ \bullet_2\ \bullet_2\ \bullet_3\ \bullet_3
> > }\_{\text{Update 2}}
> > \hspace{5pt} | \hspace{5pt}
> > \underbrace{
> >   \bullet_3\ \bullet_3\ \bullet_3\ \bullet_3\ \bullet_4\ \bullet_4
> > }\_{\text{Update 3}}
> > \hspace{5pt} | \hspace{5pt}
> > \underbrace{
> >   \bullet_4\ \bullet_4\ \bullet_4\ \bullet_4
> > }\_{\text{Update 4}}
> > $$
> >
> >
> >
> >
> > We highlight that, if boundaries are unknown, there is a chance that each update step would involve a few examples of a subsequent task (e.g., Updates 1, 2, 3). Otherwise, we would need to know exactly where each task finishes so that the data from the update does not involve a subsequent task.
> >
> > This streaming setup makes the learnability/plasticity problem harder (as the learner now has to learn from a chunk that is under distributional shift), while slightly alleviating the memory problem (as a particular task might appear in two subsequent updates). This "alleviation" is what we meant by replaying tasks into different chunks, and we apologize for not being clear. It is also worth mentioning that **this is the same setup used by the work of Zeno et. al.** [1] (see Section 6.1), and we only included the memory and architecture restrictions (which we described in Appendix J) on top. We thank the reviewer for bringing this work to our attention in their review, which served as inspiration for this new evaluation setup.
> >
> >
> > Finally, we highlight that, following the protocol of continual learning settings, **the training data is thrown away after each update** (besides the coresets, which are limited as described in Appendix J), so we are not replaying the training data for multiples tasks and strictly follows the same standards as in the other benchmarks employed in this work and prior literature.
> >
> > We hope this clarifies our previous message, and let us know if you have any further concerns. We appreciate your time and contributions while reviewing our work.
> >
> >
> > **References**
> >
> > [1] Zeno et al., Task-Agnostic Continual Learning Using Online Variational Bayes with Fixed-Point Updates. Neural Computation, 2021

---

> > > ### Comment · Reviewer_xMRf · 2025-08-05
> > >
> > > I thank the authors for the detailed and thoughtful clarification.
> > > The authors have fully addressed my main concerns.
> > > Given these clarifications and improvements, I am updating my score from 3 to 5.

---

### Official Review · Reviewer_aVHF · 2025-07-03

**Clarity:** 3
**Significance:** 3
**Originality:** 4
**Rating:** 5
**Confidence:** 4

**Summary:**

This paper looks at integrating temporal-difference style approach for limiting compounding errors in continual learning. The motivation behind this work mainly follows the bayesian framework, where instead of just considered the parameters of just the previous timestep, one should consider a combination of n previous time steps. Part of the proposed approach was inspired by the concept of TD lambda return, which is used as the variational distribution q. The authors evaluated their approach using variants of MNIST, CIFAR100 and Tiny ImageNet benchmarks. The results show that their approach had more success in mitigating catastrophic forgetting.

**Questions:**

1. It seems that the main focus is on catastrophic forgetting. However, in continual learning, catastrophic interference is also a challenge. Can the authors share how their work can or will relate to catastrophic interference?

2. Will the model also work for reinforcement learning problems?

**Ethical Concerns:**

["NO or VERY MINOR ethics concerns only"]

**Final Justification:**

The authors provided further clarification and additional empirical results which showed that their approach does indeed learn well without task knowledge.

**Limitations:**

Yes

**Quality:**

3

**Strengths And Weaknesses:**

# Strength
1. The writing is clear and the mathematical theorems used are easy to follow.
2. Figure 2 is important to give a clear motivation of this paper.
3.  I appreciate the authors providing the intuition in section C and D in the Appendix on the connection of TD targets in TD-VCL and RL. It is a creative approach of using bootstrapping targets.
4. The authors also included implementation details and hyper parameters tuned.


# Weakness
1. The baseline models are not the most competitive baseline models. For example, one model that the authors should consider which is relevant to the Bayesian continual learning literatures is the elastic weight consolidation model (EWC) [1].

2. The proposed approach requires knowledge of task information and task boundary. Are there ways to overcome this limitation?


[1] Kirkpatrick, James, et al. "Overcoming catastrophic forgetting in neural networks." Proceedings of the national academy of sciences 114.13 (2017): 3521-3526.

---

> ### Author Rebuttal · Authors · 2025-07-30
>
> Thank you for your review! We appreciate that you found our work **clear and easy to follow**, with good illustrations and implementation details, and **enjoyed the connections with RL methods and found our idea creative**. We are also happy that you understand our work as having **good quality, clarity, and significance**, and **excellent originality.**
>
> We aim to address your concerns below:
>
> **Q7** Are the current Bayesian CL baselines competitive?
>
> **A7** We start by highlighting that our work has a particular goal of **advancing Bayesian methods for CL**. We highlight that the Bayesian framework follows a principled approach that **allows the development of epistemic uncertainty-aware models**, which is crucial for robust, safe Machine Learning. These models are naturally capable of performing Active Learning [1], Out-of-distribution Detection [2], and Risk-Sentitive Optimization [3]. They also provide a new layer of interpretability for predictions via uncertainty estimates. **These are capabilities that non-Bayesian methods do not provide** and are very relevant for online/continual learning under distributional shifts and beyond.
>
> Besides that, **most methods explore design choices that are orthogonal to ours** (e.g., architecture, memory, regularization). **Given the flexibility of our objective, it can be directly combined with different methods (and even objectives)**. To validate this property, in Table 3, we extend other methods (UCL, UCB) to employ the TD objective, presenting consistent improvements. **We believe most methods would also be complementary to our objective.**.
>
> Given our scientific goal of validating the proposed learning objective under the Bayesian CL setting, we believe our baselines are representative and provide solid evidence to support our claim. And we also believe they are competitive in this particular setting. For instance, **EWC (suggested in the review) is consistently outperformed by VCL [4] (Figs. 2 and 5), UCL [5] (Figs. 3, 6, and 7), and UCB [6] (Table 2), across different benchmarks and implementations**. We believe this evidence is strong to support that our baselines outperform EWC and are, in general, strong enough for the Bayesian CL setting.
>
> **Q8** Does TD-VCL assume knowledge of task boundaries?
>
> **A8** Thank you for raising this concern. We plan to clarify it here and incorporate this clarification as part of our paper.
>
> We argue that the **TD-VCL objective (and VCL objectives in general) does not require knowledge of task boundaries**, and we provide theoretical and empirical evidence for that.
>
>
> The theoretical argument comes from the principle that the **Bayesian framework is self-consistent**: given a stream of data, the final posterior distribution should be the same regardless of how many Bayesian updates are executed.
>
> Based on that, the key thing is to realize that **the number of updates does not need to be equal to the number of tasks**. Mathematically, suppose we have a stream of $T$ tasks (represented by $t$). At a particular update $k$, we may consider a Bayesian update that includes data from multiple ($m$) sequential tasks (e.g., from $t_{a}$ to $t_{a + m}$): $\mathcal{D}\_k = \bigcup\_{j=0}^{m} \mathcal{D}\_{j}$. Crucially, this does not impose any assumptions on boundaries. Rather, once we decide where to start and end the data stream for the Bayesian update, there could be potentially many tasks included.
>
> Under the same assumptions stated in Section 3, we have that: $p(\mathcal{D}\_k \mid \theta) = \prod\_{j=0}^{m} p(\mathcal{D}\_j \mid \theta)$. And, the recursive relationship (Equation 1) also follows: $p(\theta \mid \mathcal{D}\_{1:k}) \propto p(\theta \mid \mathcal{D}\_{1:k-1}) \prod\_{j=0}^{m} p(\mathcal{D}\_{j} \mid \theta)$. Finally, following the same variational objective and ELBO derivation, we arrive at $\mathcal{L}^k(\theta) = \mathbb{E}\_{\theta \sim q\_k(\theta)}\left[ \sum\_{j=0}^{m} \log p(\mathcal{D}\_j \mid \theta) \right] - \mathcal{D}\_{\text{KL}}(q\_k(\theta) \,\|\, q\_{k-1}(\theta))$, which is a tractable objective equivalent to Eq. 3 where the number of steps $k$ is isolated from the number of tasks $T$.
>
> **Therefore, the objective itself does not discriminate or require task boundaries**. TD-VCL will estimate the likelihood terms for multiple terms simultaneously, which is something already done while replaying past tasks.
>
> Now, we highlight that most **benchmarks** do isolate tasks, including the ones in this work, which makes it convenient to consider one Bayesian update per task. We also acknowledge that our writing in the Preliminaries section leads to concluding that the method requires task boundaries by conflating the meaning of the index $t$, and **we will clarify this point in the potential camera-ready version**.
>
> To provide further practical evidence that the method does not require knowledge of boundaries, we present a new benchmark called **StreamingPermutedMNIST-Hard**.  **This benchmark does not provide any boundary between tasks.** From the full data stream of $T$ tasks, we create random chunks of sequential data and provide them to the methods for continual learning. We execute an evaluation after the complete data stream, considering held-out splits composed of all tasks. We report the average accuracy across them (equivalent to the t = 10 column in Tables 2 and 3 of the paper). The Table below shows the empirical results over 10 seeds. **We observe no negative impact in the VCL/TD-VCL methods in comparison with PermutedMNIST-Hard.** In fact, some methods improved performance, because we are likely replaying the same task into different chunks, alleviating the catastrophic forgetting challenge. The proposed objectives still outperform all other methods.
>
> **StreamingPermutedMNIST-Hard:**
>
> | Method          | t = 10         |
> |------------------|----------------|
> | Online MLE       | 0.54 ± 0.09     |
> | Batch MLE        | 0.64 ± 0.09     |
> | VCL              | 0.82 ± 0.05     |
> | VCL CoreSet      | 0.85 ± 0.04     |
> | N-Step TD-VCL    | 0.89 ± 0.02     |
> | TD(λ)-VCL        | 0.89 ± 0.02     |
>
>
>
> **Q9** Does TD-VCL assume knowledge of task information?
>
> **A9** Following the answer to the previous question, we argue that the TD-VCL objective does not require task information. We have a single replay buffer for all tasks, and the data is mixed. We also make sure that the newly introduced benchmarks (PermutedMNIST-Hard, SplitMNIST-Hard, SplitNotMNIST-Hard) are all built with single-head classification architectures.
>
> **Q10** How does TD-VCL relate to/address "catastrophic interference"?
>
> **A10** We found the formal definition of catastrophic interference and catastrophic forgetting to be equivalent, as in [7, 8]. We believe the question relates to **negative transfer between tasks**, i.e., learning a new task negatively correlates with learning another one.
>
> Such task interference is often associated with **representational incompatibility**: the features learned to solve task A may not align with useful representations for task B. Therefore, feature reusing becomes misleading, which affects performance in negatively correlated tasks. We argue that this representational incompatibility is due to learning tasks in isolation: when solely learning task A, the optimization disregards task B. Alternatively, if we have a learning signal from both tasks simultaneously, then this issue is alleviated.
>
> In this sense, we believe that the TD-VCL objective addresses such interference by allowing multi-task learning via a replay buffer that naturally emerges in its design. By replaying past tasks, the objective ensures that the learned representation does not completely disregard them, even if the buffer is tiny and limited.
>
> On the other side, if this interference is caused by "gradient interference" [9], then it would require employing gradient projection methods (as in [9]), which is out of the scope of this work but can be readily integrated into the optimization procedure.
>
>
> **Q11** Is TD-VCL applicable to RL problems?
>
> **A11** While TD-VCL is not an RL algorithm (in the sense of synthesizing policies for maximizing rewards), it is readily applicable for RL algorithms with probabilistic formulation and that require online learning. For instance, it can be applied to probabilistic modeling of dynamics [10] or value functions [11], and RL formulations that leverage sequential variational inference, such as probabilistic meta-RL [12]. Bayesian RL is indeed very appealing, as the epistemic uncertainty estimates could be directly used for active exploration or conservative policy learning.
>
> References
>
> [1] Gal et. al. Deep Bayesian Active Learning with Image Data. ICML, 2017.
>
> [2] Nguyen et. al. Out of Distribution Data Detection Using Dropout Bayesian Neural Networks. AAAI, 2022.
>
> [3] Depeweg et. al. Decomposition of Uncertainty in Bayesian Deep Learning for Efficient and Risk-sensitive Learning. ICML, 2018.
>
> [4] Nguyen et. al. Variational Continual Learning. ICLR, 2018.
>
> [5] Ahn et. al. Uncertainty-based Continual Learning with
>  Adaptive Regularization. NeurIPS, 2019.
>
> [6] Ebrahimi et. al. Uncertainty-Guided Continual Learning with Bayesian Neural Networks. ICLR, 2020.
>
> [7] McCloskey et. al. Catastrophic Interference in Connectionist Networks: The Sequential Learning Problem. Psychology of Learning and Motivation, 1989.
>
> [8] French et. al. Catastrophic interference in connectionist networks: Can it be predicted, can it be prevented?. NeurIPS, 1993.
>
> [9]Yu et. al. Gradient Surgery for Multi-Task Learning. NeurIPS, 2020.
>
> [10] Chua et. al. Deep Reinforcement Learning in a Handful of Trials using Probabilistic Dynamics Models. NeurIPS, 2018.
>
> [11] Ghavamzadeh et. al. Bayesian Reinforcement Learning: A Survey, 2015.
>
> [12] Zintgraf et. al. VariBAD: A Very Good Method for Bayes-Adaptive Deep RL via Meta-Learning. ICLR, 2020.

---

### Official Review · Reviewer_LVzR · 2025-07-07

**Clarity:** 3
**Significance:** 2
**Originality:** 3
**Rating:** 4
**Confidence:** 3

**Summary:**

- This paper studies continual learning with known task boundaries on image classification
- This paper looks at the Bayesian continual learning (particularly VCL), and reformulates the object so that it is a weighted combination of task likelihoods estimated through a small coreset, and task likelihoods estimated through Kl-divergences with past posteriors
- The authors argue that this allows better performance than VCL since it uses multiple previous posterior estimates, rather than only the most recent one
- The authors make connections to Temporal difference learning in reinforcement learning
- There is a variation for VCL, UCB and UCL
- Empirically the TD variant of the algorithm outperforms the vanilla variant in all scenarios

**Questions:**

- How does this compare to non-bayesian methods of continual learning? As the authors stated, generally this method can be adapted to any regularization based approach, so it would be interesting to see more methods
- Is is possible to limit the number of posteriors which need to be stored? As stated in the weaknesses section, currently we need to store N posteriors. Is it possible to do a reweighting trick similar to what was done to derive the TD objective to limit the number of past posteriors which need to be stored?

**Ethical Concerns:**

["NO or VERY MINOR ethics concerns only"]

**Final Justification:**

I will maintain the score of 4, as there are clear contributions of the paper, but the scope and potential impact of the paper seem more limited, namely due to the more limited evaluation of the method compared to stronger continual learning methods.

**Limitations:**

See Weaknesses/Questions Section

**Paper Formatting Concerns:**

None.

**Quality:**

3

**Strengths And Weaknesses:**

Strengths:
- Well motivated and theoretically principled
- Well written
- Interesting connections to reinforcement learning which is valuable to the community
- Consistent improvement over non-TD baselines

Weaknesses
- Comparisons are only done relatively to the non-TD variants (VCL, UCB, and UCL). While improvement is consistent, these methods are generally not competitive CL methods, and there is no comparison to newer higher performance methods
- This method requires a large number of additional storage. Namely, it requires the use of a coreset to estimate the monte-carlo objective, and also requires storing the model posterior estimate for all past objectives. Compared to VCL, this is large, since it requires only storing the previous posterior estimate. Over large N this could be problematic (as storage space grows linearly with N)

---

> ### Author Rebuttal · Authors · 2025-07-30
>
> Thank you for your review! We appreciate that you found our work **theoretically principled and well motivated, well-written**, with **consistent results** and with **interesting connections to RL**. We also appreciate that you found our work to have good quality, clarity, and originality.
>
> We aim to address your concerns below:
>
> **Q4** Why compare with only Bayesian CL baselines? How would this method compare with other families of CL methods?
>
> **A4** We agree that providing a more exhaustive set of CL baselines would provide a better perspective on the current CL research landscape. However, CL research spans several directions, adopting different assumptions and desiderata. We opted not to broaden the scope too much, as there are literally hundreds of different methods available. Otherwise, it would be really hard to control the experiments and perform fair comparisons. Alternatively, our work keeps baselines consistent in these terms, allowing us to make direct claims about the **impact of the proposed objective**, which is our object of study.
>
> Additionally, **most methods explore orthogonal design choices** (e.g., architecture, memory, regularization). **Given the flexibility of our objective, it can be directly combined with different methods (and even objectives)**. To validate this property, in Table 3, we extend other methods (UCL, UCB) to employ the TD objective, presenting consistent improvements. **We believe most methods would also be complementary to our objective.**.
>
> Lastly, as stated in the paper, our work has a particular goal of **advancing Bayesian methods for CL**. We highlight that the Bayesian framework follows a principled approach that **allows the development of epistemic uncertainty-aware models**, which is crucial for robust, safe Machine Learning. These models are naturally capable of performing Active Learning [1], Out-of-distribution Detection [2], and Risk-Sentitive Optimization [3]. They also provide a new layer of interpretability for predictions via uncertainty estimates. **These are capabilities that non-Bayesian methods do not provide** and are very relevant for online/continual learning under distributional shifts and beyond.
>
>
> **Q5** Memory Cost Analysis and Discussion
>
> **A5** We first analyze the memory cost associated with TD-VCL. The buffer size does present memory complexity of O($n$), where $n$ is the n-Step hyperparameter. This cost is lower than or equal to the core sets in VCL CoreSet. Nonetheless, the designed benchmarks purposely limit the size of the replay buffer to contain up to 200 data points for previously observed tasks, which is negligible given the full training data of the current task (60,000 data points). Thus, it does not configure a major bottleneck.
>
> The posteriors' regularization also presents memory complexity of O($n$). For smaller networks (e.g., the ones used for MNIST and NotMNIST benchmarks), the memory usage is comparable: 200 MNIST data points use ~0.60 MB, while the posterior uses ~0.68 MB (assuming float32). Naturally, the posteriors use more memory as we leverage larger/deeper networks.
>
> As described in the Limitations paragraph (Section 6), we do acknowledge that TD-VCL may increase the memory requirements. Nonetheless, we would like to highlight why this is not necessarily a major limitation and also present alternatives to optimize memory usage.
>
> 1) **Memory Complexity is a function of $n$, which the user controls**. The buffer size and number of posteriors are defined by $n$. If memory is a bottleneck, one can control $n$ to satisfy memory constraints. **Crucially, $n$ is not always equal to the number of tasks.** Our robustness analysis (Appendix L) shows that performance increases monotonically with $n$ up to a level where it may saturate. Therefore, any $n > 1$ should be better than vanilla VCL, and, if performance is expected to saturate, one can also set $n$ to be much lower than the number of tasks. The employed hyperparameters (Appendix H) suggest that we can usually assume a value of $n$ that is lower than the number of tasks.
>
> 2) **Assume a memory-efficient variational family $\mathcal{Q}$**. Since memory may be a challenge for large Bayesian networks, there are some alternative architectures, such as last-layers variational methods or bayesian LoRA adapters [4, 5, 6], which approximates the posterior distribution in a fixed number of parameters, drastically reducing the required memory (at the cost of expressiveness of the variational family).
>
>
> 3) **Store previous posteriors in cheaper memory alternatives**. Since TD-VCL does NOT use previous posteriors for inference but only for computing the KL regularization, they do not need to occupy GPU memory. In fact, the regularization term could be computed asynchronously on CPU (or even with an external computer) while the GPU is used to generate predictions and estimate the likelihood terms. While implementation is more involved, it allows the use of both CPU/GPU and avoids having previous posteriors in GPU memory, which is usually the bottleneck.
>
> 4) **Estimate TD objective with fewer posteriors but covering older timesteps**. A corollary of Proposition 4.4 is that we can represent the learning target as **any** combination of n-step TD targets. This means that we may store posteriors at every $m$ steps, instead of every step. In this case, given $T$ tasks, we only store $T / m$ posteriors. Naturally, this leads to a different way of estimating the learning objective, but ensures that it is covering older tasks to prevent catastrophic forgetting.
>
>
> Lastly, we highlight that there are realistic Continual Learning settings where storing posteriors is not a major bottleneck. For instance, when **continually learning on embedded systems with access to a cloud storage**. These embedded systems (mobile phones, wearables) usually present limited storage/GPU memory onboard. Some problems require on-the-fly model adaptation and, for privacy-related reasons, the data must be kept on the device for a limited time. Nonetheless, we may upload model snapshots to the cloud without problems (sometimes this upload is required to conduct quality evaluation/audits). A concrete example is an on-device speech recognition model on a smartwatch adapting to a user's voice. We believe our TD-VCL objective is well-suited for this problem setting.
>
>
> **Q6** Is it possible to limit the number of posteriors that need to be stored?
>
> **A6** Yes. As we stated in Q5, one could:
> - Control $n$ to use fewer posteriors and satisfy memory requirements (as we highlighted, $n$ is a hyperparameter and not necessarily equal to the number of previously observed tasks); and
> - Use fewer posteriors from a longer history of tasks by maintaining posteriors only after every $m$ timesteps.
>
>
> References
>
> [1] Gal et. al. Deep Bayesian Active Learning with Image Data. ICML, 2017.
>
> [2] Nguyen et. al. Out of Distribution Data Detection Using Dropout Bayesian Neural Networks. AAAI, 2022.
>
> [3] Depeweg et. al. Decomposition of Uncertainty in Bayesian Deep Learning for Efficient and Risk-sensitive Learning. ICML, 2018.
>
> [4] Snoek et. al. Scalable Bayesian Optimization Using Deep Neural Networks. ICML, 2015.
>
> [5] Melo et. al. Deep Bayesian Active Learning for Preference Modeling in Large Language Models. NeurIPS, 2024.
>
> [6] Yang et. al. Bayesian Low-rank Adaptation for Large Language Models. ICLR, 2024.

---

> ### Comment · Area_Chair_DBeX · 2025-08-07
>
> Hi Reviewer LVzR,
>
> This is a kind reminder to review the authors’ rebuttals and check whether your concerns have been adequately addressed.
>
> If clarification is needed, we encourage you to engage in further discussion before the end of discussion phase.
>
> AC

---

> ### Comment · Reviewer_LVzR · 2025-08-08
>
> Thanks for the detailed responses to my questions. I think the contributions of this paper are fairly clear, however they are somewhat limited in scope (namely being mainly a contribution to the Bayesian CL literature, which itself is a narrow part of continual learning).  I will maintain the positive score (4), but because of the limited evaluation compared to stronger (non-bayesian) CL baselines, it is unclear how impact this paper would be for the larger CL community, so I will not vouch for a significantly stronger acceptance.

---

### Official Review · Reviewer_Fkvs · 2025-07-12

**Clarity:** 4
**Significance:** 3
**Originality:** 3
**Rating:** 5
**Confidence:** 3

**Summary:**

The paper proposes Temporal-Difference Variational Continual Learning（TD-VCL）, a novel variational continual learning (CL) objective that mitigates catastrophic forgetting by integrating multiple past posterior estimates through *n*-step KL regularization and temporal-difference (TD) targets. This paper's presentation is good and clear, which makes it easy to understand and follow. The conclusion is aligned with its experiment results, and the author discussed their limitations in the last section.

**Questions:**

Please see Weakness

**Ethical Concerns:**

["NO or VERY MINOR ethics concerns only"]

**Final Justification:**

I appreciate the authors' detailed response, which has resolved the issues I raised. In light of these clarifications, I will increase my rating.

**Limitations:**

Please see Weakness

**Quality:**

4

**Strengths And Weaknesses:**

## Strength:
1. The investigated problem: continue learning is important.

2. The core idea of leveraging multiple past posteriors (via n-step KL and TD(λ)-VCL) to reduce compounding approximation errors is theoretically proved and innovative.

3. The presentation is clean and clear. I really like the organization of figures 3 and tables 1,2,3.

4. The appendix is detailed, making the experiments more convincing.

5. The experiment results are good, which outperforms the baseline: MLE series and vanilla VCL and align with the authors' conclusion.

## Weakness:
1. I notice sometimes TD(λ)-VCL outperforms n-Step TD-VCL, but sometimes fall behind (table1, SplitMNIST-Hard, t=5; table1, SplitNotMNIST-Hard, t=4). So what do you like the reason?  When should we use n-Step TD-VCL and when to use TD(λ)-VCL?

2. As all the experiments in this work is MLE, AlexNet architectures,  I'm very curious about the performance on transformer architecture models, as today's LLMs frequently include this architecture, discussing TD-VCL performance on transformers will bring much greater impact.

3. The computation cost is also important. I'd appreciate it if this analysis is included.

---

> ### Author Rebuttal · Authors · 2025-07-30
>
> Thank you for your review! We appreciate that you found our work **theoretically principled and innovative**, our **presentation clean/clear**, the experiments **detailed and convincing**, with **good results**. We also appreciate that you found our quality/clarity **excellent**, and significance and originality **good**.
>
> We aim to address your concerns below:
>
> **Q1** Does sometimes TD(λ)-VCL fall behind N-Step TD-VCL? When should we use one or the other?
>
> **A1** First, we would like to highlight that in both examples raised by the reviewer, given the overlap in confidence intervals, we cannot conclude with high statistical confidence that one method outperforms the other -- they are statistically on par. Nonetheless, the question remains valid and important.
>
> TD(λ)-VCL is a generalization of N-Step TD-VCL. As we presented in Appendix E, TD(λ)-VCL forms a spectrum of CL algorithms, and we recover N-Step TD-VCL when $\lambda \rightarrow 1$. Therefore, the "choice" depends on λ, which controls how much the learning objective should prioritize recent posteriors. If one believes that the most recent posterior retains the knowledge of previous tasks, then a higher λ should work better. Otherwise, one should use lower values as past estimates contain information that has not propagated over the recursive updates. In practice, it depends on the continual learning problem and the potential transfer/interference among tasks. The recommendation is to start from TD(λ)-VCL and tune the λ hyperparameter.
>
> **Q2** TD-VCL in transformer architectures
>
> **A2** The choice of architecture follows prior work in the area of Bayesian CL [1-4], so that we can establish a fair and reliable comparison with past claims from this literature. We believe this choice is solid as it reflects what is advisable to fit the tasks in the employed benchmarks.
>
> Furthermore, **our proposed objective is agnostic to the architectural inductive biases as long as it approximates a variational distribution**. In this sense, our method is readily applicable to **Bayesian transformers, which has already been shown to be effective for Variational Inference [5]**. Since we employ the same mechanism, we believe the effectiveness also extends to our problem setting.
>
> **Q3** Computational cost
>
> **A3** Thank you for your suggestion. We provide a detailed analysis below, which will be incorporated in a new appendix of our work.
>
> We divide this analysis into three perspectives:
>
>
>
> 1) Training Cost: The training cost is defined by how we compute Equations 4/5, which relies on two terms: MC estimation of the likelihood term (I) and the KL regularization terms (II). (I) is the standard cross-entropy loss for classification, but averaged across different $\theta$ sampled from the variational distribution. **We approximate this average with a single sample**, so the cost is the same as for standard classification with MLE objective. (II) is the KL regularization term, which is computed in closed form. It is lightweight since it does not rely on the data/forward passes in the network. Comparatively, the costs of VCL and TD-VCL are approximately the same, as (I) (the bottleneck) is roughly equivalent. We also implement early stopping, which greatly reduces the number of training epochs, reducing computational resources considerably in comparison with the implementations of prior work.
>
> 2) Inference Cost: The Bayesian inference involves approximating the posterior predictive distribution $p(y^* \mid \mathbf{x}^*, \mathcal{D}\_{1:t}) = \mathbb{E}\_{q\_t(\theta)} [p(y^\* \mid \theta, \mathbf{x}^\*, \mathcal{D}\_{1:t})]$ via MC sampling, which again depends on the number of samples of $\theta$. In our work, **we used a single sample** to be computationally fair across all baselines. Thus, our cost is simply a forward pass in the network, as in a standard classifier. More samples would incur additional computational cost, but it should also lead to higher performance.
>
>
> 3) Hyperparameter search cost: This cost can be controlled by fixing the search compute. As described in Appendix G, we limit all methods (including baselines) to at most 1 GPU day budget. Our method incorporates 2 hyperparameters on top of VCL (n, λ), and results in Appendix L presents a moderate-to-good robustness to the choices, which suggests this cost could be cut down if required.
>
>
> References:
>
> [1] Nguyen et. al. Variational Continual Learning. ICLR, 2018.
>
> [2] Ahn et. al. Uncertainty-based Continual Learning with  Adaptive Regularization. NeurIPS, 2019.
>
> [3] Ebrahimi et. al. Uncertainty-Guided Continual Learning with Bayesian Neural Networks. ICLR, 2020.
>
> [4] Bayesian Adaptation of Network Depth and Width for Continual Learning. ICML, 2024.
>
> [5] Sankararaman et al. BayesFormer: Transformer with Uncertainty Estimation, 2022.

---

> ### Comment · Area_Chair_DBeX · 2025-08-07
>
> Hi Reviewer Fkvs,
>
> This is a kind reminder to review the authors’ rebuttals and check whether your concerns have been adequately addressed.
>
> If clarification is needed, we encourage you to engage in further discussion before the end of discussion phase.
>
> AC

---

### Decision · Program_Chairs · 2025-09-17

**Decision:**

Accept (poster)

**Comment:**

This paper proposes a variational continual learning method inspired by temporal-difference updates. The work is theoretically sound, clearly presented, and demonstrates consistent improvements over VCL and related Bayesian baselines. Some concerns remain about limited evaluation, scalability, comparisons to non-Bayesian methods, and the potential experiments on other scaled settings. The rebuttal addressed key questions and strengthened confidence in the approach. Overall, the contribution is solid within Bayesian continual learning. The reviewers converged toward positive ratings. Taking the overall strengths and weaknesses into account, the AC recommends acceptance.